# The effect of inhibition of receptor tyrosine kinase AXL on DNA damage response in ovarian cancer

Xun Hui Yeo [1,2], Vignesh Sundararajan[2], Zhengwei Wu [1,2], Zi Jin Cheryl Phua[1], Yin Ying Ho [3], Kai Lay Esther Peh [3], Yi-Chia Chiu[4], Tuan Zea Tan [2], Dennis Kappei [2,5,6], Ying Swan Ho [3], David Shao Peng Tan[2,7], Wai Leong Tam [1,2,5,6,8,11] & Ruby Yun-Ju Huang [4,9,10,11 ✉]

AXL is a receptor tyrosine kinase that is often overexpressed in cancers. It contributes to pathophysiology in cancer progression and therapeutic resistance, making it an emerging therapeutic target. The first-in-class AXL inhibitor bemcentinib (R428/BGB324) has been granted fast track designation by the U.S. Food and Drug Administration (FDA) in STK11-mutated advanced metastatic non-small cell lung cancer and was also reported to show selective sensitivity towards ovarian cancers (OC) with a Mesenchymal molecular subtype. In this study, we further explored AXL's role in mediating DNA damage responses by using OC as a disease model. AXL inhibition using R428 resulted in the increase of DNA damage with the concurrent upregulation of DNA damage response signalling molecules. Furthermore, AXL inhibition rendered cells more sensitive to the inhibition of ATR, a crucial mediator for replication stress. Combinatory use of AXL and ATR inhibitors showed additive effects in OC. Through SILAC co-immunoprecipitation mass spectrometry, we identified a novel binding partner of AXL, SAM68, whose loss in OC cells harboured phenotypes in DNA damage responses similar to AXL inhibition. In addition, AXL- and SAM68-deficiency or R428 treatment induced elevated levels of cholesterol and upregulated genes in the cholesterol biosynthesis pathway. There might be a protective role of cholesterol in shielding cancer cells against DNA damage induced by AXL inhibition or SMA68 deficiency.

[1] Genome Institute of Singapore (GIS), Agency for Science, Technology and Research (A*STAR), 60 Biopolis Street, Genome, Singapore 138672, Republic of Singapore. [2] Cancer Science Institute of Singapore, National University of Singapore, 14 Medical Drive, Singapore 117599, Republic of Singapore. [3] Bioprocessing Technology Institute (BTI), Agency for Science, Technology and Research (A*STAR), 20 Biopolis Way, Centros, Singapore 138668, Republic of Singapore. [4] Graduate Institute of Oncology, College of Medicine, National Taiwan University, Taipei, Taiwan. [5] Department of Biochemistry, Yong Loo Lin School of Medicine, National University of Singapore, 10 Medical Drive, Singapore 117597, Republic of Singapore. [6] NUS Center for Cancer Research, Yong Loo Lin School of Medicine, National University of Singapore, Singapore, Republic of Singapore. [7] Department of Haematology-Oncology, National University Cancer Institute, Singapore, Republic of Singapore. [8] School of Biological Sciences, Nanyang Technological University, 60 Nanyang Drive, Singapore 637551, Republic of Singapore. [9] School of Medicine, College of Medicine, National Taiwan University, Taipei, Taiwan. [10] Department of Obstetrics & Gynaecology, Yong Loo Lin School of Medicine, National University of Singapore, Singapore, Republic of Singapore. [11] The authors jointly supervised this work: Wai Leong Tam, Ruby Yun-Ju Huang. ✉email: rubyhuang@ntu.edu.tw

AXL is a receptor tyrosine kinase (RTK) belonging to the TAM (Tyro3, AXL, Mer) family. It is overexpressed in multiple cancers such as breast, lung, and ovarian cancer (OC) with the expression levels correlating with cancer prognosis[1,2]. AXL has also been shown to contribute to tumour progression in metastasis and the acquisition of drug resistance via the activation of pathways such as MAPK/Erk and PI3K/Akt signalling[2–5]. Inhibition of AXL using small molecule inhibitors or silencing AXL via short-hairpin knockdown systems was able to prevent tumour progression and to re-sensitise treatment-resistant cancer cells[6–9], making it an emerging therapeutic target[2,10]. Recently, the first-in-class AXL inhibitor bemcentinib (R428/BGB324) has been granted fast-track designation by the U.S. Food and Drug Administration (FDA) in STK11-mutated advanced metastatic non-small cell lung cancer (NSCLC)[11]. Clinical trials focusing on combination regimes between AXL inhibition and chemotherapeutic agents, or immunotherapies are being launched. Therefore, uncovering novel combination strategies with AXL remains critical.

In OC, studies have shown that the overexpression of AXL could be found in patients who developed resistance to platinum-based chemotherapy not responding to second-line treatment[12–14]. Previously, we have identified AXL being highly expressed in the mesenchymal (Mes)-subtype ovarian tumour tissue and cancer cell lines[1]. The Mes subtype conferred poor clinical survival outcomes among the 5 gene expression molecular subtypes[15]. AXL inhibition by using R428 attenuated the expressions of phosphorylated AXL (pAXL) and the downstream phosphorylated Erk (pErk), significantly impaired cell motility and suppressed *in ovo* tumour formation[1,15]. AXL inhibition by using a decoy aptamer has also been shown to induce DNA damage in OC[13]. The new emerging role of AXL associated with DNA damage, DNA damage response (DDR), and homologous recombination (HR) was further shown in NSCLC, triple-negative breast cancer (TNBC), and head and neck squamous cell carcinoma (HNSCC)[16]. Therefore, AXL inhibition might enable the sensitisation of cells to the PARP1 inhibitors[16]. With the DDR and HR pathway now being considered as the most crucial pathway for synthetic lethality in OC, it is therefore plausible that the combination targeting both AXL and the DDR pathways might be effective.

Here, we explored the mechanisms encompassing the role of AXL in DNA damage and DDR in OC, aiming to provide novel findings that will pave way to the rational use of AXL and DDR inhibitors in combination therapy. Our results demonstrated that AXL inhibition by (R428/BGB324) or the loss of AXL expression induced DNA damage through replication stress. R428 treatment further sensitised OC cells to ATR inhibitors. A novel AXL protein binding partner SAM68 was identified by using stable isotope labelling by amino acids in cell culture (SILAC) mass spectrometry. Full-length and cleaved AXL in the nuclear fractions were found to translocate into the nucleus. The loss of AXL and the loss of SAM68 shared similar molecular and functional consequences in the increases in DNA damage and upregulating genes involved in the cholesterol biosynthesis pathway. SAM68-deficient OC cells showed enhanced sensitivity to the combination of AXL and ATR inhibition. Furthermore, AXL- and SAM68-deficient or R428-treated cancer cells had elevated levels of cholesterol, indicating a protective cellular metabolic response against AXL inhibition-induced DNA damage.

## Results

**AXL inhibition with small molecule inhibitor R428 decreases cell proliferation and induces G2/M arrest.** To explore molecular alterations upon AXL inhibition, global transcriptomic profiling was performed using SKOV3, a Mes-subtype OC cell line expressing high AXL levels (Supplementary Fig. 1a). Among differentially expressed genes and pathways, genes associated with cell cycle, mitosis, and DNA replication were downregulated upon AXL inhibition (Fig. 1a). Functional validation carried out using the Mes-subtype cell lines SKOV3 and HeyA8 demonstrated a decrease in cell proliferation after R428 treatment (Fig. 1b, Supplementary Fig. 1b). Strikingly, cell cycle analysis exhibited an initial increase in S phase followed by $G_2/M$ phase upon AXL inhibition, signifying a $G_2/M$ phase arrest (Fig. 1c, d, Supplementary Fig. 1c, d). In addition, GSEA analysis of the down-regulated genes showed the enrichment in the DNA repair pathway (Supplementary Fig. 1e), suggesting a plausible increase in DNA damage upon AXL inhibition. Since cell cycle arrest often occurs in response to DNA-damaging agents[17,18], the activation of $G_2/M$ phase cell cycle arrest and the decrease in cell proliferation upon AXL inhibition might be the consequence of the induction of DNA damage.

**AXL inhibition and knockdown increase DNA damage.** Immunofluorescence (IF) staining of DNA damage markers showed a significant increase in the fluorescence intensity of γH2AX (Fig. 2a, b) and pRPA32 (Fig. 2c) upon R428 treatment in both SKOV3 and HeyA8, suggesting that AXL inhibition increased DNA damage and induced replication stress. These changes were also validated using the AXL knockdown system (shAXL) (Supplementary Fig. 2a–g). The increase in 53BP1 foci seemed to be cell-line dependent (Fig. 2d).

Consistent with the IF staining results, the activation of DDR markers following AXL inhibition showed elevated levels of γH2AX and pRPA32 expression (Fig. 2e, Supplementary Fig. 2h). Increased expressions of pATM, pATR and pCHK1 were observed at early time points (Fig. 2e) while pCHK2 levels increased at much later time points (Fig. 2e, Supplementary Fig. 2h). This indicated that upon AXL inhibition, DDR first responded to the presence of single-stranded DNA breaks (SSB), followed by the presence of double-stranded DNA breaks (DSB) which might be caused by collapsed stalled replication forks during replication stress[19]. Collectively, AXL inhibition might lead to SSB followed by DSB accompanied by the activation of DDR proteins.

**AXL inhibition sensitises cells to ATR inhibitors.** Knowing that AXL inhibition induced the increase in ssDNA and replication stress, we explore the possibility of a combination treatment using the AXL inhibitor R428 with an ATR inhibitor BAY1895344. ATR is a serine/threonine-kinase that senses the presence of ssDNA at stalled replication forks and activates DNA damage checkpoint leading to cell cycle arrest[20]. We hypothesised that the combination of ATR and AXL inhibition would induce DNA damage to catastrophic levels, thus eventually leading to cell death. Combination treatment was carried out in a panel of OC cell lines harbouring different gene expression profiles with varying AXL expression levels (Supplementary Fig. 3a), using their respective 20% inhibitory concentration ($IC_{20}$) of R428 (Supplementary Fig. 3b–h) and varying doses of BAY1895344. An overall reduction in the $IC_{50}$ of BAY1895344 was observed in the Mes-subtype cell lines—SKOV3 (Fig. 3a), HeyA8 (Fig. 3b), RMG5 (Fig. 3c) and OVTOKO (Supplementary Fig. 3j)—showing a greater shift in the dose response curve of BAY1895344 compared to the non-Mes cell lines—PEO1 (Fig. 3d), OVCA429 (Supplementary Fig. 3k) and CH1 (Supplementary Fig. 3l). In addition, Mes-subtype cell lines also harboured an overall lower average $IC_{50}$ of BAY1895344 when treated in combination (Supplementary Fig. 3i). Combination

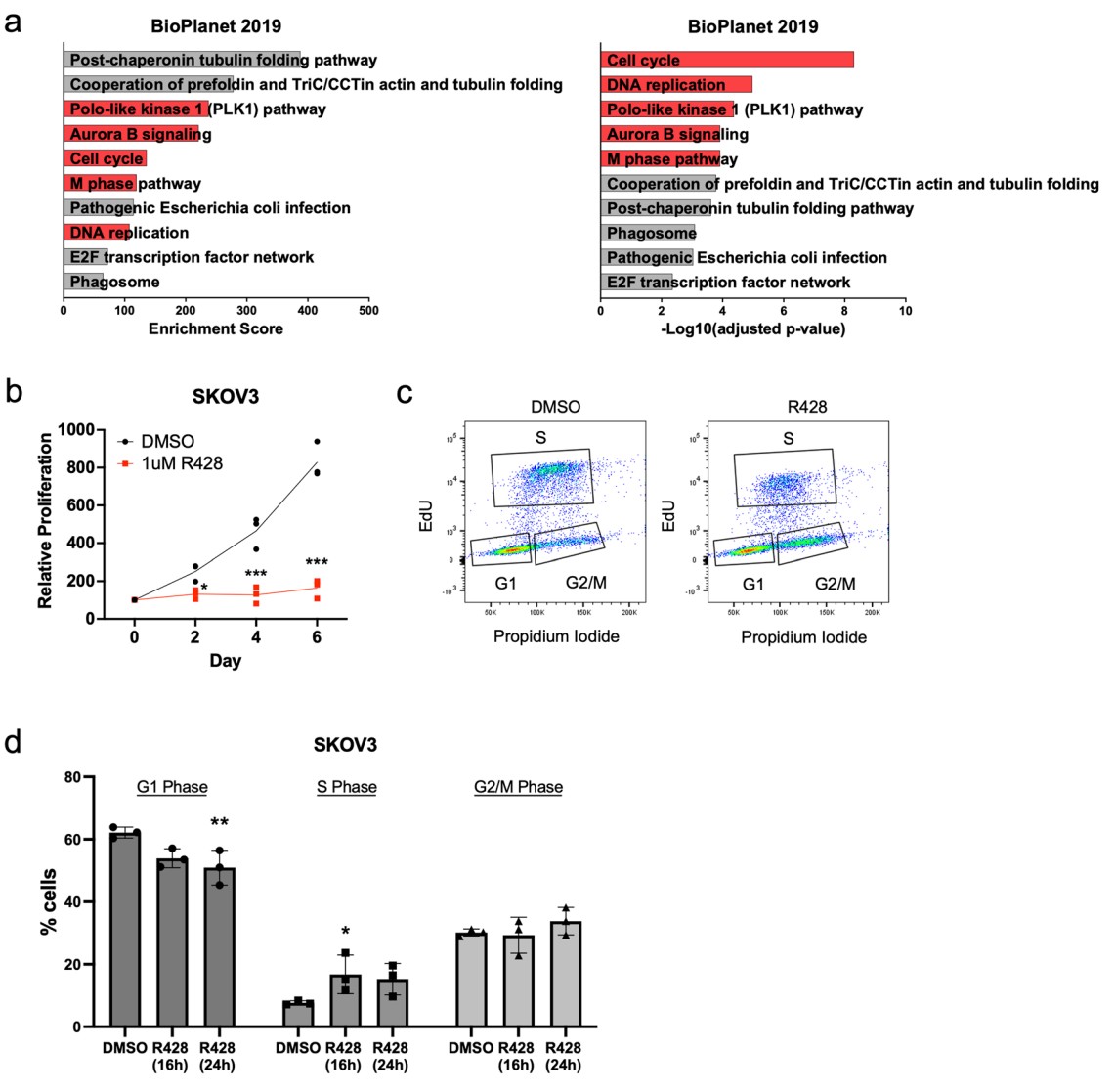

**Fig. 1 AXL inhibition associated with downregulation of cell cycle, mitosis and DNA replication. a** Top ten downregulated pathways upon AXL inhibition. Pathways associated with the downregulation of the cell cycle, mitosis and DNA replication are highlighted in red. **b** Proliferation curve of SKOV3 treated with R428 compared to DMSO. Cell viability was normalised against mean viability at day 0 assessed with CellTiter-Glo. **c** Representative flow cytometry plot of cell cycle analysis in SKOV3 treated with R428 for 24 h compared to DMSO. Ten thousand events were recorded for each sample. **d** Quantification of cell cycle analysis of SKOV3 treated with R428 for 16 h and 24 h showing the percentage of cells in different phases of the cell cycle ($G_1$, S, $G_2$/M [$G_2$ and mitosis]). Data represent mean ± s.e.m.; *$p < 0.05$, **$p < 0.01$, ***$p < 0.001$, ****$p < 0.0001$, determined by Dunnett's multiple comparisons tests.

index (CI) calculated using the Chou–Talalay method[21] showed either synergism (CI < 1) or additive effect (CI = 1) for all cell lines (Fig. 3e) with a favourable dose reduction index (DRI > 1) (Fig. 3f). Moreover, fluorescence intensity and immunoblotting of γH2AX in SKOV3 (Fig. 3g, Supplementary Fig. 3m, n) and RMG5 (Fig. 3h, Supplementary Fig. 3m) were significantly elevated in the combination treatment compared to either single agent, suggesting that inhibition of both AXL and ATR further induced DNA damage. Validation using another ATR inhibitor VE-821 also yielded similar results (Supplementary Fig. 3o–w). Collectively, these data showed that AXL inhibition was able to sensitise cells to ATR inhibition by further increasing DNA damage. To determine its translatability, SKOV3 was treated *in ovo* using the chick chorioallantoic membrane (CAM) xenograft assay. The tumour growth in CAM was greatly suppressed when treated with the AXL and ATR combination treatment compared to either single agent (Fig. 3i).

Interestingly, low AXL-expressing PEO1 also responded well to the combination treatment (Fig. 3d). PEO1 is a well-known OC cell line derived from a *BRCA2* mutated high-grade serous ovarian carcinoma (HGSOC) patient[22,23]. This raised the question if HR deficiency could be the basis for another combination treatment with AXL inhibition. Contrary to what was observed, only *BRCA* mutant cell lines (Fig. 4a, b) and two out of five *BRCA* wild-type cell lines (Fig. 4c, d, Supplementary Fig. 4a–c) responded to combination treatment with R428 and Olaparib, a PARP1 inhibitor. *BRCA2* mutant PEO1 showed a synergistic effect while *BRCA1/2* mutant RMG5 showed an increasing additive effect (CI = 1) (Fig. 4e) with favourable DRI (Fig. 4f). However, an antagonistic effect (CI > 1) was observed in all cell lines with *BRCA* wild-type status (Fig. 4e). Validation using another PARP1 inhibitor, Niraparib, also exhibited similar results (Supplementary Fig. 4d–h). As *BRCA* mutants have defective HR repair, DNA damage induced by AXL inhibition and accumulated by PARP1 inhibition would not be resolved, therefore contributing to cell death. In addition, the failure of *BRCA* wild-type cell lines to respond to combination treatment suggested that with the presence of functional HR repair, the

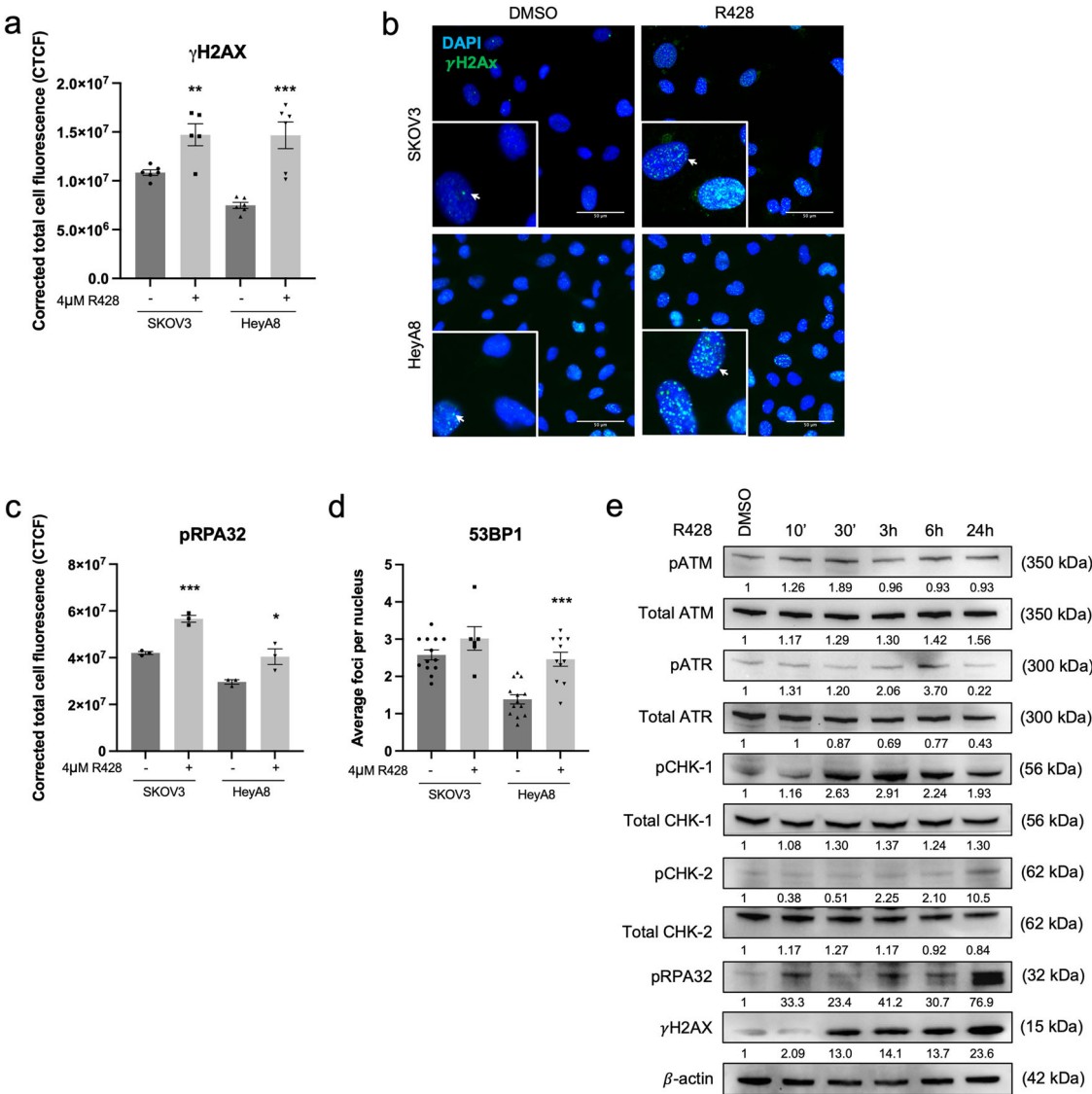

**Fig. 2 AXL inhibition increases DNA damage and DNA damage response pathways. a** Fluorescence intensity quantification of γH2AX in DMSO Vs R428 treated in SKOV3 and HeyA8. Data represent mean ± s.e.m.; *$p < 0.05$, **$p < 0.01$, ***$p < 0.001$, determined by two-tailed $t$-test with statistical significance with Welch's correction. **b** Immunofluorescence staining of γH2AX staining (arrows) in DMSO and R428 treated SKOV3 and HeyA8. Blue, DAPI; Green, γH2AX. **c** Fluorescence intensity quantification of pRPA32 in DMSO Vs R428 treated SKOV3 and HeyA8. **d** Nuclear foci quantification of 53BP1 in DMSO Vs R428 treated SKOV3 and HeyA8. Data in **c** and **d** represent mean ± s.e.m.; *$p < 0.05$, **$p < 0.01$, ***$p < 0.001$, determined by two-tailed $t$-test with statistical significance with Welch's correction. **e** Immunoblot of DDR markers upon AXL inhibition at the indicated time points in SKOV3. Numbers below blots reflect protein band intensity normalised against DMSO.

extent of DNA damage induced by AXL inhibition was still within the tolerable threshold of the cell.

**SAM68 is a novel binding partner to AXL**. Like the EGFR family of RTK, AXL is known to exist as a nuclear form[8,24]. To elucidate the possible mechanism of AXL-mediated DNA damage, we explored potential novel binding partners of AXL by using SILAC co-immunoprecipitation (Co-IP) mass spectrometry (Fig. 5a). Among the pulldown proteins in the mass spectrometry analysis, AXL was also identified. Potential binding partners of AXL were filtered based on the differential SILAC ratio. The protein Src associated in mitosis, of 68 kDa (SAM68/KHDRBS1) was among the highest SILAC ratio. SAM68 is a KH domain RNA-binding protein that plays a variety of roles in cellular processes including RNA stability and nuclear export[25–28]. Emerging evidence shows that SAM68 has critical functions in

DDR[29] and governs PARP1 activation at DNA damage sites triggered by DSBs, and poly(adenosine diphosphate [ADP]-ribosyl)ation (pADPr) of PARP1 activation during DNA damage[26]. The interaction of SAM68 with AXL was validated using the nuclear fraction of high AXL-expressing OC (SKOV3 and HeyA8) and NSCLC (H1299) lines. The association between AXL and SAM68 was detected only in the nuclear fractions but not the cytoplasmic fractions (Fig. 5b), confirming that SAM68 was a novel binding partner of AXL in the nucleus and this interaction was not specific to OC.

**The role of AXL-SAM68 interaction in protecting against DNA damage**. We moved on to determine whether AXL inhibition might alter SAM68 expression or the protein interaction. In both R428-treated SKOV3 and HeyA8 cells, there was no change in SAM68 abundance in either the nuclear or cytoplasmic fraction

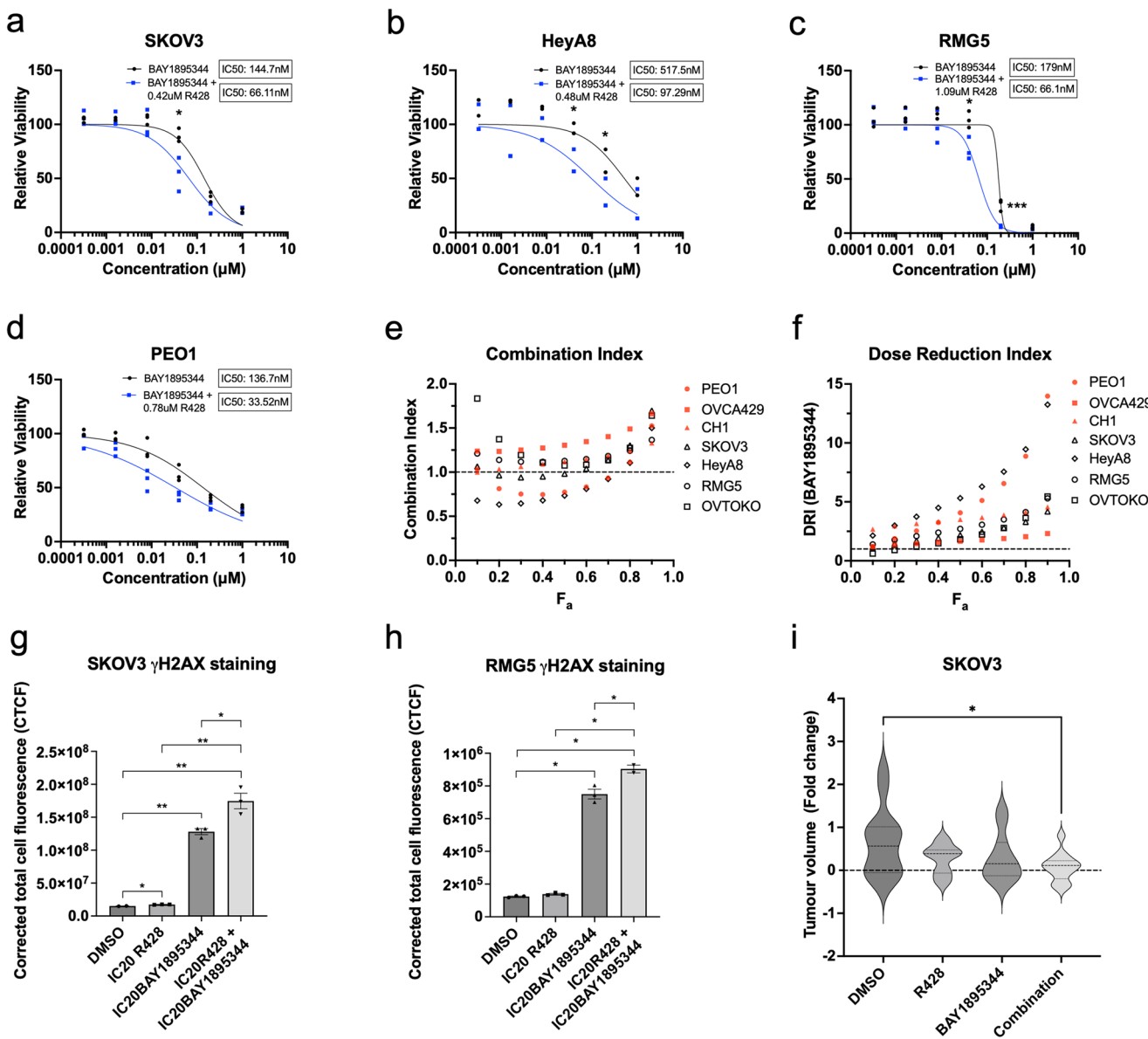

**Fig. 3 AXL inhibition sensitises cells to ATR inhibition.** Combination treatment of R428 with BAY1895344 in **a** SKOV3, **b** HeyA8, **c** RMG5 and **d** PEO1. **e** Combination index of R428 and BAY1895344. **f** Dose reduction index of BAY1895344; Fa fraction affected. **g** Fluorescence intensity quantification of γH2AX in SKOV3 and **h** RMG5 treated with combination treatment of AXL and ATR inhibition. Data represent mean ± s.e.m.; *$p < 0.05$, **$p < 0.01$, ***$p < 0.001$, determined by two-tailed $t$-test with statistical significance with Welch's correction. **i** Relative tumour volume change against the baseline. Data represent mean ± s.e.m.; *$p < 0.05$, **$p < 0.01$, ***$p < 0.001$, determined by two-tailed $t$-test with statistical significance with Welch's correction; $n = 14$ tumours treated with DMSO, $n = 11$ tumours treated with R428, $n = 11$ tumours treated with BAY1895344, and $n = 13$ treated with combination treatment.

(Fig. 5c). Similarly, the interaction between AXL and SAM68 in the pull-down was not affected (Fig. 5d). Interestingly, an increased nuclear AXL abundance was observed in both R428-treated OC cell lines (Fig. 5c). A cleaved version of AXL (~50 kDa) was detected only in the cytoplasmic fraction before R428 treatment, but it further increased in levels in both the cytoplasmic and nuclear fractions upon AXL inhibition (Fig. 5c). This indicated that AXL inhibition might increase AXL proteolysis and nuclear abundance of both full-length and cleaved AXL.

Next, to determine whether the loss of its binding partner would interfere with DDR, CRISPR–Cas9 mediated AXL and SAM68 knockout (KO) clones were generated. Increased γH2AX levels were observed in both AXL- and SAM68-KO clones (Fig. 6a, b). Elevated levels of pADPr expression were also observed in the AXL KO clones (Fig. 6a) with similar findings

exhibited in one HeyA8 SAM68 KO clone (Fig. 6b). The data suggested that DNA damage was enhanced when either AXL or SAM68 was depleted. We next determined if the DNA damage induced by AXL inhibition would be abrogated upon SAM68 depletion (Fig. 6c). The expression of pADPr and γH2AX increased in the R428-treated SAM68 control clone. However, the increase in γH2AX induced by R428 treatment was still present in the SAM68 KO clones with the further increase in pADPr expression (Fig. 6c), suggesting that the increase in DNA damage mediated by AXL inhibition was not affected by the depletion of SAM68. In addition, to show that the increase in pADPr expression was due to the activation by PARP1 in response to increased DNA damage, R428-treated cells were further treated with Olaparib (Fig. 6c). The increase of pADPr was clearly inhibited by the addition of Olaparib, suggesting that the

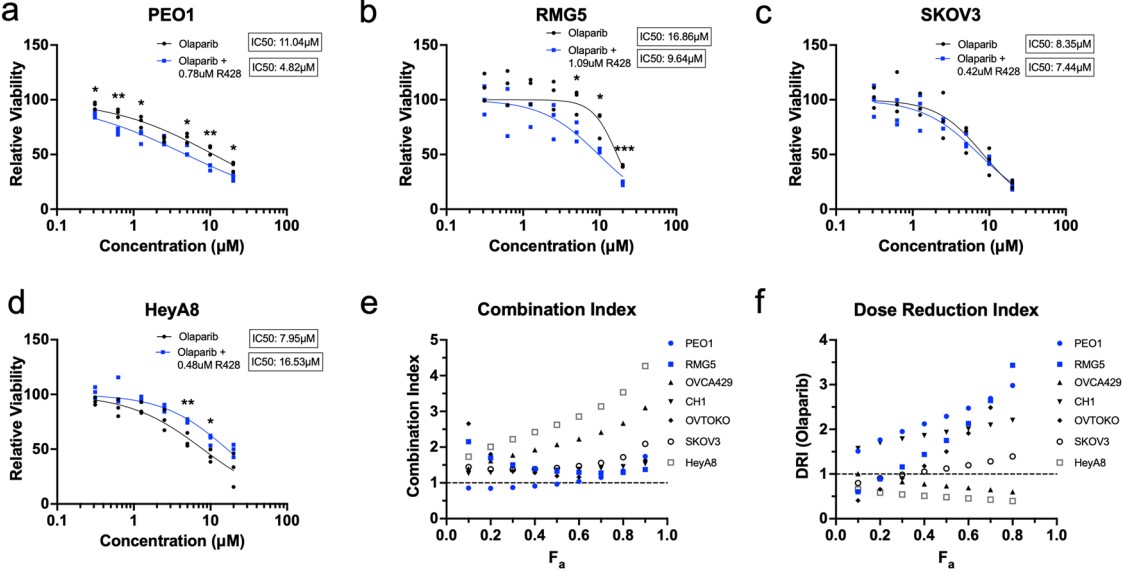

**Fig. 4 AXL inhibition sensitises *BRCA* mutant cells to PARP1 inhibition.** Combination treatment of R428 with Olaparib in **a** *BRCA2* mutant PEO1, **b** *BRCA1/2* mutant RMG5, **c** SKOV3 and **d** HeyA8. **e** Combination index of R428 and Olaparib. **f** Dose reduction index of Olaparib. Fa fraction affected.

induction of pADPr mediated by AXL inhibition might also be independent of SAM68.

As the loss of SAM68 did not affect R428-induced DNA damage, we wondered if cleaved AXL and/or nuclear AXL would be affected by SAM68 depletion. Prior to AXL inhibition, both full-length AXL (130 kDa) and cleaved AXL (~50 kDa) were detected in both the cytoplasmic and nuclear fractions of both SAM68 control and KO cell lines (Fig. 6d). Interestingly, loss of SAM68 further increased the expression of cleaved AXL in the nuclear fraction (Fig. 6d). This suggested that the presence of nuclear AXL might not require SAM68 but the loss of SAM68 might further contribute to nuclear AXL, which was similar to the effect observed in the R428-treated parental cell lines (Fig. 5c). Upon R428 treatment, this increase in full-length and cleaved AXL was still observed in SAM68-deficient cell lines (Fig. 6d). This further confirmed that the nuclear pool of AXL was not depleted upon the loss of SAM68 but was significantly increased after AXL inhibition.

We moved on to determine if the loss of SAM68 could further escalate DNA damage caused by the combination treatment of AXL and ATR inhibitors (Fig. 6e, f). Consistently, AXL inhibition was able to sensitise control cell lines to ATR inhibition. Strikingly, the loss of SAM68 caused a greater shift in the dose–response curve, causing increased sensitisation to the combination treatment. Furthermore, CI also showed a synergistic effect with favourable DRI (Fig. 6g, h). Collectively, our results suggested that SAM68, being the novel binding partner of AXL, might have a parallel role in DNA damage and DDR. The DNA damage induced by AXL inhibition might be independent of SAM68 and the loss of SAM68 itself could trigger DNA damage and DDR, and further sensitise cells to the combination treatment using AXL and ATR inhibitors.

**AXL or SAM68 loss-of-function upregulates cholesterol biosynthesis**. To explore the functions that might be regulated by AXL and SAM68, transcriptomic analysis of AXL KO and SAM68 KO cell lines was done to explore commonly altered pathways. Pathway analysis[30,31] revealed that 'cholesterol biosynthesis' was enriched in the 857 commonly up-regulated genes in both AXL and SAM68 deficient cells (Fig. 7a, b). This

observation was also consistent with the transcriptomic analysis in the R428-treated SKOV3 cell line (Supplementary Fig. 5a, b). Genes related to the cholesterol biosynthesis pathway (*SREBF2*, *ACAT2*, *HMGCS1*, *HMGCR*, *FDFT1*) were validated, and their gene expression levels were upregulated upon AXL inhibition in both SKOV3 and HeyA8 (Fig. 7c, d) as well as AXL and SAM68 deficient cells (Supplementary Fig. 5c, d). The time- and dose-dependent changes of these cholesterol biosynthesis genes upon AXL inhibition (Supplementary Fig. 5e–h) further revealed the dynamic regulation where their expression peaked at 24 h after R428 treatment in both SKOV3 and HeyA8. A similar trend was also observed in protein expression upon AXL inhibition (Fig. 7f, Supplementary Fig. 5i). To assess if these findings were context-specific, transcriptomic analysis was also performed in an NSCLC cell line A549 and a normal human mammary epithelial line MCF10a. Increased expression of cholesterol biosynthesis genes were observed in A549 (Supplementary Fig. 5j–l) upon AXL inhibition, but not in MCF10a (Supplementary Fig. 5m, n). This indicated that the upregulation of cholesterol biosynthesis genes upon AXL inhibition might not be specific to OC and might not occur in normal cells. Further validation of the expression changes of these cholesterol biosynthesis genes upon combination treatment with AXL and ATR inhibitors was also conducted in SKOV3. Upregulation of cholesterol biosynthesis genes was also observed in mono-treatment of the ATR inhibitor and in combination treatment (Supplementary Fig. 5o).

To determine if cholesterol levels would correlate with the differential regulation of cholesterol biosynthesis pathway genes, cell lines were stained with filipin III[32]. Filipin III staining mainly localised at the cell membrane in DMSO-treated cells, while upon AXL inhibition, filipin III staining mainly localised in the cytoplasm and in the perinuclear region (Fig. 7e, Supplementary Fig. 5p). This observation was still present 72 h post R428 treatment, suggesting the presence of cholesterol even after downregulation of cholesterol biosynthesis genes. Total intracellular cholesterol quantified also showed a sustained increase 72 h post R428 treatment (Supplementary Fig. 5q–s). Consistent with the transcriptomic data (Supplementary Fig. 5m, n), upregulation of cholesterol was not observed upon AXL inhibition in MCF10a (Supplementary Fig. 5p, t). To further confirm this observation, liquid

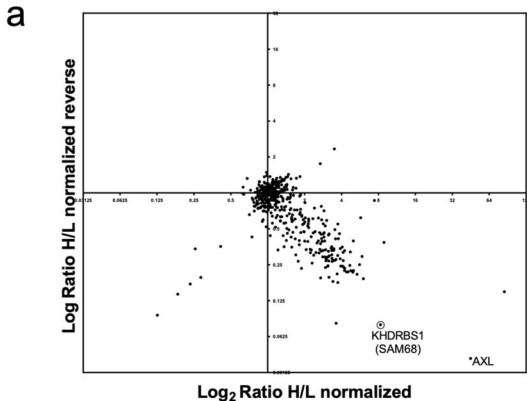

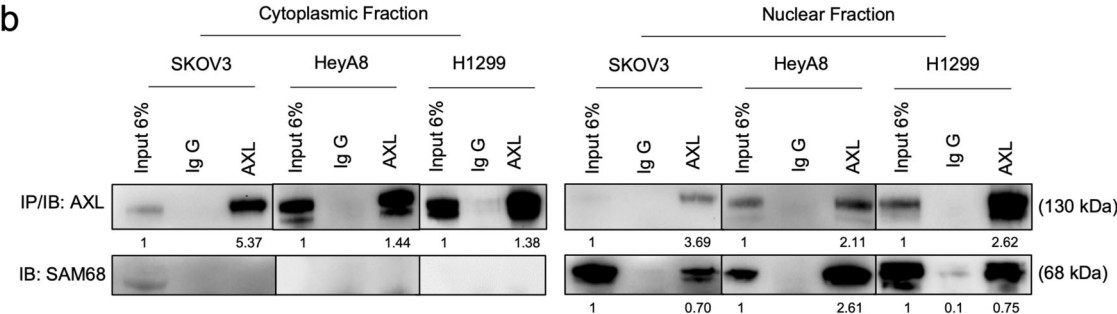

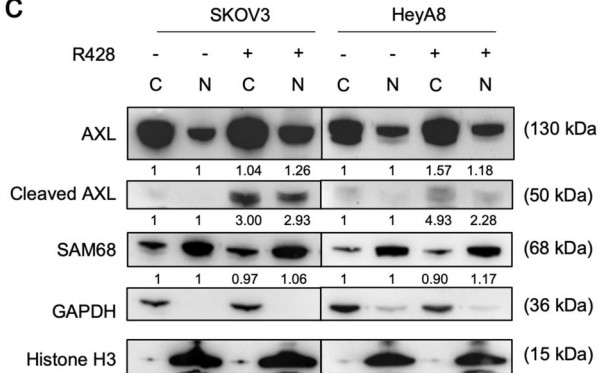

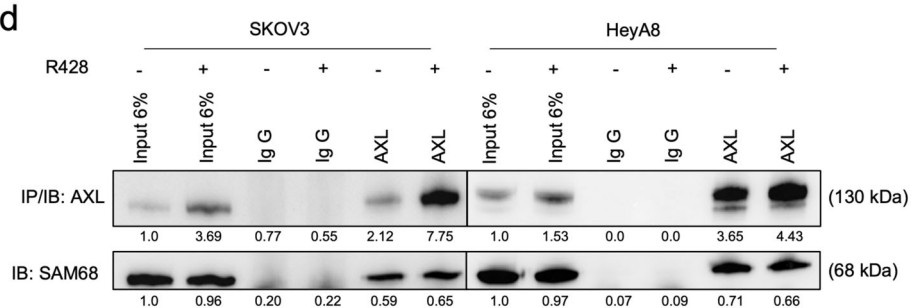

**Fig. 5 Novel nuclear AXL binding partner SAM68. a** Two-dimensional normalised plot of SILAC Co-IP in OVCA429. Circled was a potential target chosen for validation. **b** Co-IP of AXL using a cytoplasmic and nuclear fraction of SKOV3, HeyA8 and H1299 cell lines. **c** Immunoblot showed no changes in SAM68 levels and increased AXL nuclear translocation upon AXL inhibition in SKOV3 and HeyA8. **d** Nuclear Co-IP of SAM68 upon AXL inhibition in SKOV3 and HeyA8; Numbers below blots reflect protein band intensity normalised against respective control; C cytoplasmic fraction, N nuclear fraction, IB immunoblot, IP co-immunoprecipitation.

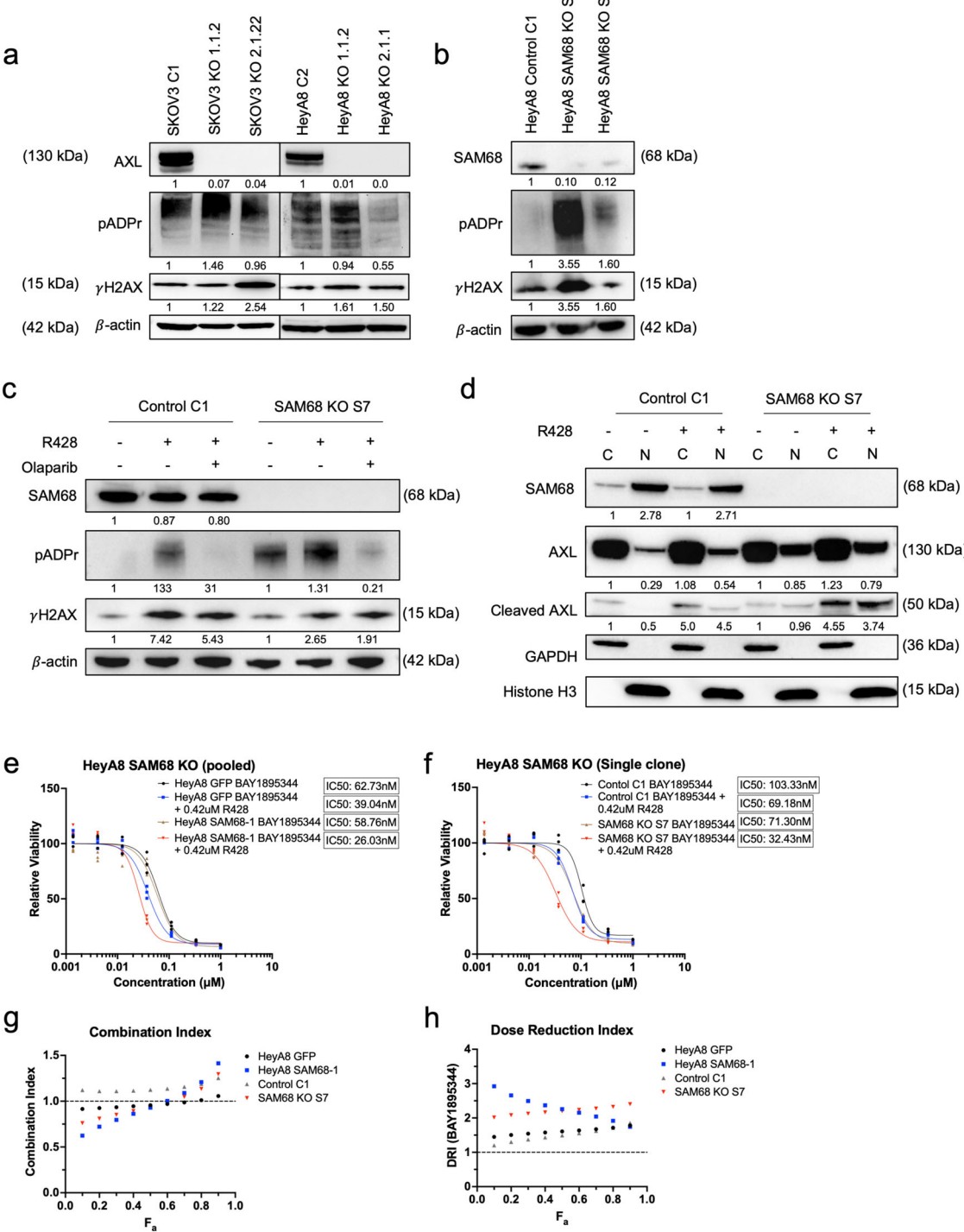

**Fig. 6 Depletion of SAM or AXL increases DNA damage and pADPr. a** CRISPR-Cas9 mediated AXL knockout in SKOV3 and HeyA8. **b** CRISPR-Cas9 mediated SAM68 knockout in HeyA8. **c** Immunoblot of HeyA8 SAM68 KO treated with R428 and Olaparib. **d** Immunoblot of AXL nuclear localisation upon AXL inhibition in HeyA8 SAM68 KO. Numbers below blots reflect protein band intensity normalised against respective control. **e** Combination treatment of R428 with ATR inhibitor BAY1895344 in pooled SAM68 KO cell lines and **f** single clone SAM68 KO cell lines. **g** Combination index of R428 and BAY1895344. **h** Dose reduction index of BAY1895344. $F_a$ Fraction affected.

chromatography–mass spectrometry (LC–MS) was also performed in R428-treated SKOV3 where both free cholesterol and cholesteryl esters (CEs) were identified (Fig. 7g, h). A slight increase in free cholesterol while a significant increase in CEs was detected upon AXL inhibition. This observation was also further validated using Amplex™ Red Cholesterol Assay Kit (Supplementary Fig. 5u).

**Cholesterol accumulation aids in cell survival by countering DNA damage.** As AXL inhibition increased cholesterol biosynthesis, the functional effects elicited by cholesterol were further investigated. Proliferation assays revealed that the supplementation of cholesterol at varying concentrations (1, 5 and 10 µg/ml) attenuated the decrease in proliferation caused by AXL inhibition in a dose-dependent manner (Fig. 8a and

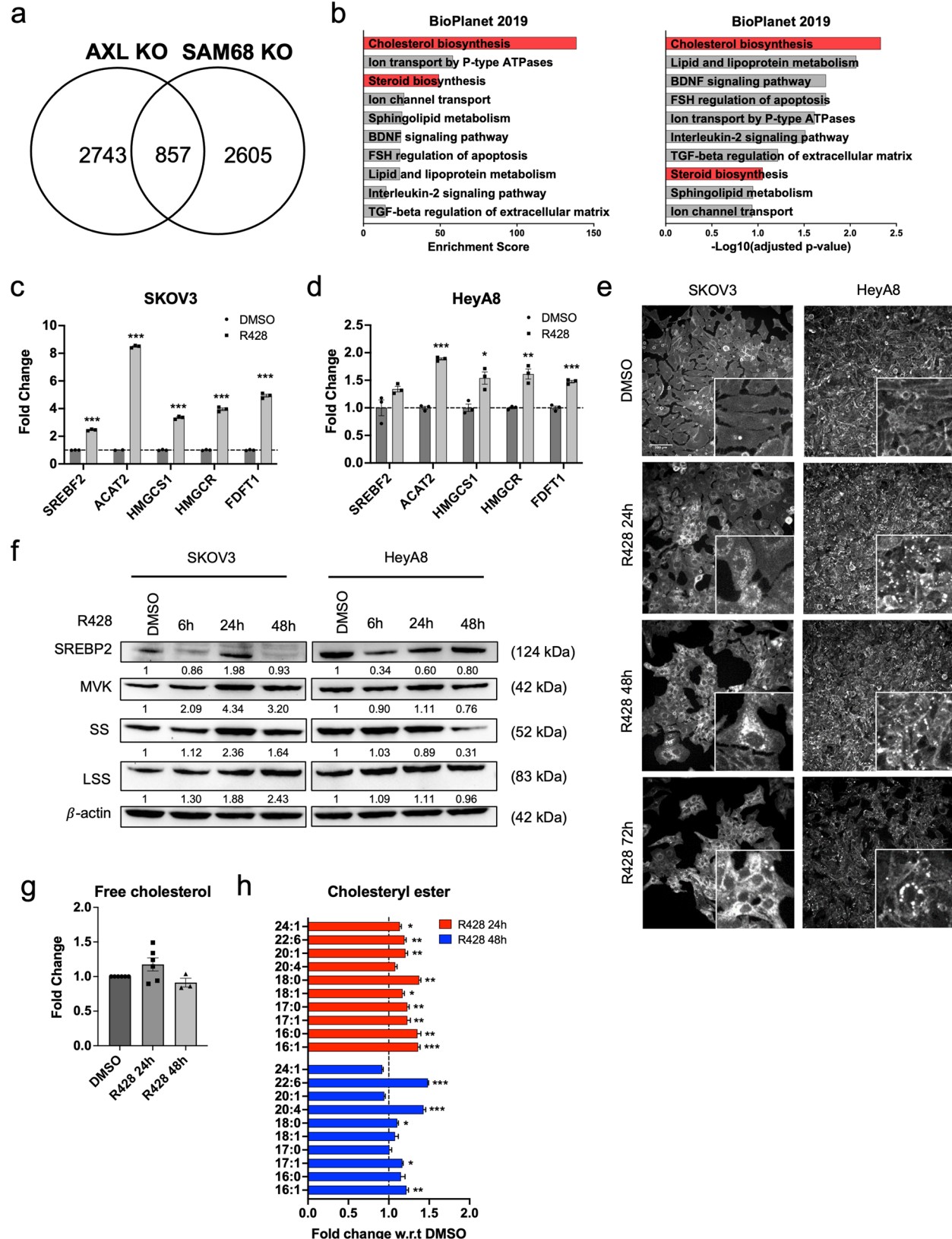

Supplementary Fig. 6a). Total intracellular cholesterol levels were also quantified to ensure that cholesterol was being taken up by the cells (Supplementary Fig. 6b). This suggested that the increase in cholesterol biosynthesis upon AXL inhibition may be a survival mechanism that counteracts AXL inhibition.

It is therefore plausible that cholesterol may prevent or decrease DNA damage induced by AXL inhibition. We further investigated if the addition of cholesterol was able to limit the DNA damage burden caused by AXL inhibition. Consistently, treatment with an AXL inhibitor showed an increase in γH2AX

**Fig. 7 AXL KO, SAM68 KO and AXL inhibition increase cholesterol biosynthesis. a** Venn diagram depicting the number of genes upregulated in HeyA8 AXL KO and SAM68 KO cell lines and their common genes upregulated. **b** Top 10 upregulated pathways of common differentially expressed genes in AXL KO and SAM68 KO cell lines. Red indicates candidate pathways chosen for validation. **c** Relative fold change in expression of cholesterol biosynthesis genes in SKOV3 and **d** HeyA8; *$p < 0.05$, **$p < 0.01$, ***$p < 0.001$, determined by two-tailed $t$-test with Welch's correction. **e** Filipin III staining in SKOV3 and HeyA8. **f** Immunoblot of cholesterol biosynthesis protein expression levels peaking at 24 h upon AXL inhibition in SKOV3 and HeyA8. The numbers below blots reflect protein band intensity normalised against DMSO. **g** Quantification of free cholesterol levels and **h** cholesteryl esters levels upon treatment with AXL inhibition using LC–MS; *$p < 0.05$, **$p < 0.01$, ***$p < 0.001$, determined by two-tailed $t$-test with Welch's correction.

accompanied by a significant increase in pADPr (Supplementary Fig. 6c). Upon the supplementation of cholesterol, a slight decrease in γH2AX and pADPr indicated the reduction of DNA damage (Supplementary Fig. 6c). Similarly, validation on HeyA8 AXL KO and SAM68 KO cell lines also showed a slight decrease in γH2AX expression upon cholesterol supplementation (Supplementary Fig. 6d, e).

Since supplementation of cholesterol reduces DNA damage induced by AXL inhibition and the loss of AXL or SAM68, upregulation of the cholesterol biosynthesis pathway might be a protective mechanism against increased DNA damage. We thus test whether the inhibition of cholesterol biosynthesis would prevent the increase in cholesterol upon AXL inhibition and potentiate DNA damage. Upon knocking down SREBF2 (Fig. 8b), downregulation of downstream cholesterol biosynthesis genes (Fig. 8c) together with lower intracellular cholesterol levels was observed (Fig. 8d). This decrease in cholesterol levels correlated with the elevated levels of γH2AX expression in shSREBF2 cell lines (Fig. 8e), which was further enhanced with AXL inhibition (~30% increase) (Fig. 8e). Importantly, supplementation of cholesterol delayed the increase in γH2AX expression induced by AXL inhibition, albeit not to the levels of DMSO. These findings were further validated by knocking down the downstream gene of SREBF2, HMGCR (Supplementary Fig. 6f–i).

## Discussion
Recent studies have unravelled the multi-faceted roles of AXL in tumorigenesis[33,34]. The role of AXL in DNA damage and DDR has been gradually coming to light[35,36]. By using OC as the disease model, we further expanded the understanding of the functional and therapeutic consequences in DDR upon AXL inhibition (Fig. 7f). Upon AXL inhibition, cell proliferation was decreased with $G_2/M$ phase cell cycle arrest in response to DNA damage with the increase in γH2AX, elevated levels of pRPA32 and classic DDR markers such as CHK1. Activation of ATR-CHK1 at stalled replication forks is known to lead to $G_2/M$ phase cell cycle arrest[37,38]. The data collectively suggested that AXL inhibition induced DNA damage possibly through perturbations of cell cycle progression causing replication stress[17,18,35]. Although our observations were in line with previous reports[13,16,36], an increase in γH2AX expression and DDR expression upon AXL inhibition was not evident in melanoma cell lines[39]. Conversely, the downregulation of DDR markers was reported in multiple cancers[16,39]. This suggested that the role of AXL in DNA damage repair might be context-dependent.

We also highlighted the potential of targeting the DDR pathway as an avenue for cancer treatment in AXL-overexpressing cancers. Pharmacological inhibition of AXL-induced DNA damage through replication stress and subsequent activation of DDR, contributing to lethality. The co-inhibition of DDR and AXL signalling created an additive effect in increasing DNA damage to further trigger apoptosis and ultimately improve treatment efficacy. The combination treatment of the AXL inhibitor R428 with ATR inhibitors exacerbated DNA damage compared to single agent treatments. In addition, a combination of AXL and PARP1 inhibitors was able to improve the

therapeutic index of BRCA1/2 mutant OC cell lines, presenting important clinical relevance as approximately 40-50% of OC patients have mutations in either BRCA1/2 genes or genes that play a role in HR[40,41]. Our study was in agreement with previous findings showing a positive effect of AXL inhibition in combination with DDR inhibitors in multiple cancers[36,39,42]. AXL inhibition was also reported to induce HR deficiency, leading to synergistic effects when combined with PARP1 inhibition in high AXL-expressing TNBC, NSCLC and HNSCC cell lines[16]. Therefore, this relationship between AXL and DDR can be exploited to improve the therapeutic index, paving the way to the rational use of AXL and DDR inhibitors in combination therapy.

Through SILAC Co-IP mass spectrometry, we discovered a novel AXL nuclear binding partner SAM68. The interaction of AXL and SAM68 existed exclusively in the nucleus. We found the loss of either AXL or SAM68 increased DNA damage and elicited similar DDRs. It thus seemed that AXL's function against DNA damage might be independent of SAM68. However, loss of SAM68 could further enhance sensitivity toward the combination treatment of AXL and ATR inhibition, demonstrating the additive effects of targeting multiple DNA damage or DDR pathways.

Findings on AXL cleavage and nuclear abundance in our OC cell lines were also consistent with previous reports showing proteolytic cleavage of AXL and nuclear translocation of cleaved AXL across a variety of cancer cell lines[24]. This is unsurprising as translocation of RTKs into the nucleus has been widely reported[43]. Interestingly, we observed increased nuclear abundance of full-length and cleaved AXL upon AXL inhibition. While the reason for the translocation of different AXL forms is unclear, it is plausible that there are differential roles between full-length and cleaved AXL in the nucleus. We also found that the loss of SAM68 further enhanced the nuclear abundance of cleaved AXL. The data suggested that the loss of SAM68 and AXL inhibition caused similar effects in increasing AXL cleavage and nuclear abundance of AXL, which could be a contributing factor to the increased sensitivity to the combination treatment of AXL and ATR inhibition in SAM68-deficient cell line. As the biological function of AXL-SAM68 interaction remains to be elucidated, we speculated that nuclear AXL-SAM68 interaction could be important in R-loop-associated DNA damage as SAM68 have RNA-/DNA-binding ability, or that the RNA-binding activity of SAM68 could be regulated by AXL[27,28,44].

In addition to AXL and SAM68's independent roles in DNA damage or DDR, we found that loss of either protein resulted in increased cholesterol biosynthesis with elevated cholesterol and CE levels. We showed that supplementation of cholesterol in R428-treated OC cell lines could aid in cell survival. Under physiological conditions, cholesterol accumulation is cytotoxic when cholesterol homoeostasis is dysregulated[45]. However, cancer cells exploit this and often upregulate enzymes involved in cholesterol biosynthesis to meet the demand of increased cholesterol for membrane biosynthesis and other functional needs[46–52]. Moreover, the accumulation of CEs serves as reservoirs that cancer cells can access when cholesterol is in demand[53–55]. Several studies have reported a link between cholesterol and DNA damage with inhibition of cholesterol

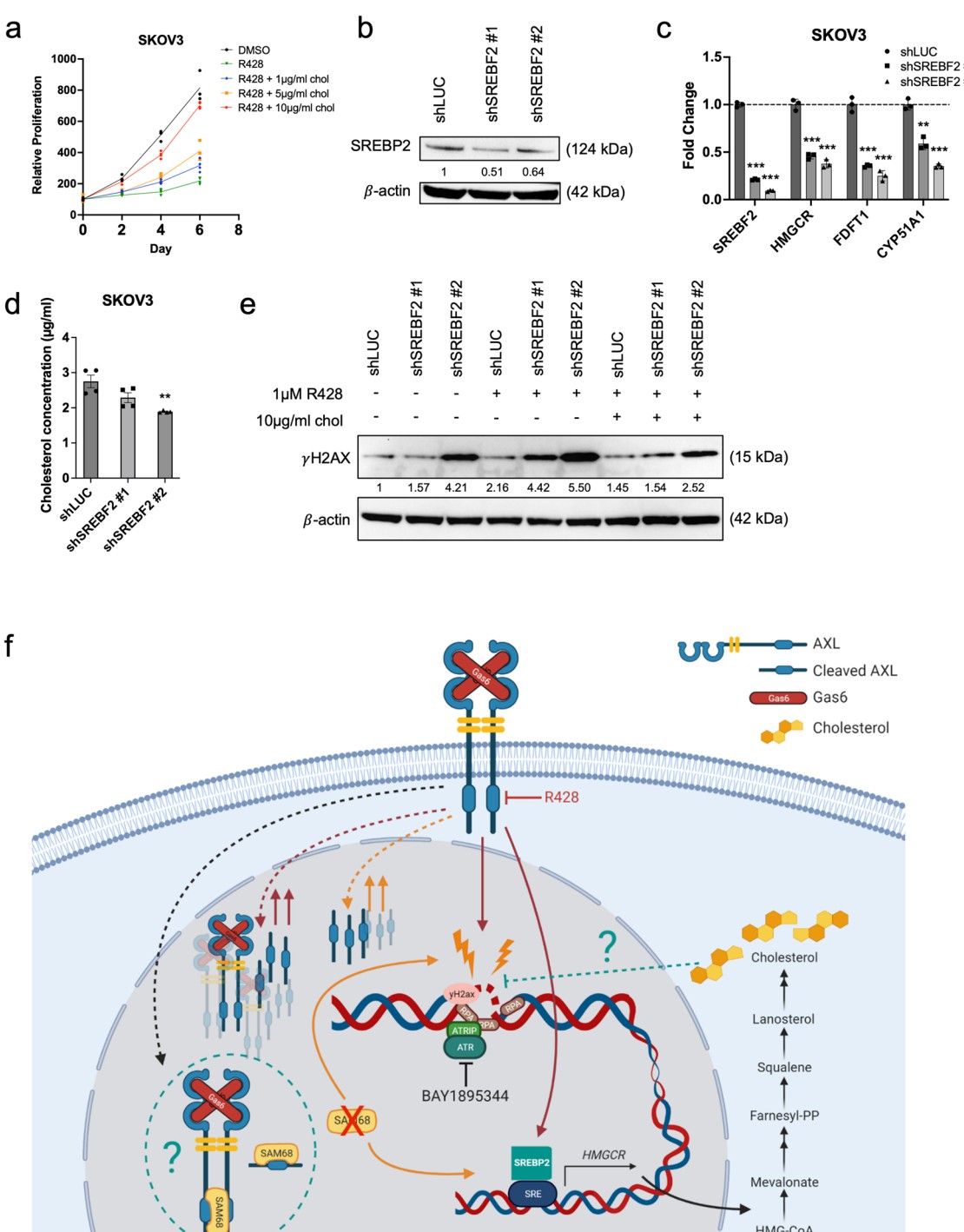

**Fig. 8 Increased cholesterol mitigates DNA damage induced by AXL inhibition. a** Proliferation curve of SKOV3 treated with R428 supplemented with cholesterol. **b** SREBF2 knockdown in SKOV3 (shSREBF2). **c** Relative expression levels of cholesterol biosynthesis genes in shSREBF2 cell lines. **d** Quantification of total cholesterol in shSREBF2 cell lines; *$p < 0.05$, **$p < 0.01$, ***$p < 0.001$, determined by two-tailed $t$-test with Welch's correction. **e** Immunoblot of shSREBF2 cell lines treated with R428 with or without supplementation of cholesterol for 3 days. Numbers below blots reflect protein band intensity normalised against shLUC; Chol cholesterol. **f** Working model illustrating the role of AXL in DNA damage and DDR, and the associated therapeutic vulnerabilities. Red continuous lines indicate the mechanism of AXL inhibition.; Red dotted lines indicate an unknown mechanism upon AXL inhibition. Orange continuous lines denote phenotype upon loss of SAM68. Orange dotted lines indicate an unknown mechanism upon loss of SAM68. Green dotted line indicates an unknown mechanism. The image was generated using BioRender.

biosynthesis suppressing DDR molecules, thus attenuating DNA repair[56–59]. In our study, suppressing cholesterol biosynthesis using short-hairpin knockdown systems to deplete the key cholesterol biosynthesis genes increased DNA damage. Supplementation of cholesterol in these cholesterol biosynthesis

deficient cell lines decreased AXL inhibition-induced DNA damage, suggesting a possible role of cholesterol in mitigating DNA damage. Non-human primate studies reported that a high cholesterol diet increased cell proliferation[58] and decreased DNA damage in the heart[57], indicating that cholesterol might be acting

as an in vivo antioxidant, providing a protective effect against DNA damage. Moreover, in vitro works have also shown that derivatives of cholesterol such as bile acids and oxidised cholesterol can inhibit DNA breakage[60]. Although these studies are beyond the context of cancer, they nonetheless support the proposed protective role of cholesterol against DNA damage. Increased cholesterol could also protect cells against DNA damage through cholesterol-mediated drug efflux[61] or through modulating signalling pathways[62]. As the protective role of cholesterol against DNA damage is not widely investigated, elucidating the mechanism of how cholesterol may limit DNA damage is essential. With cholesterol appearing to be important in preventing DNA damage and aiding in cell survival, potential combination treatment with inhibitors of the cholesterol biosynthesis pathway can be exploited to better improve current therapeutic treatments.

## Methods

**Cell lines and culture conditions**. All media and supplements were purchased from Gibco unless otherwise stated. OC cell line SKOV3, OVCA429, CH1 and NSCLC cell line H1299 and A549 were maintained using Dulbecco's Modified Eagle Medium (DMEM) media supplemented with 10% (v/v) foetal bovine serum (FBS). OC cell lines HeyA8, PEO1, OVTOKO and RMG5 were maintained using Roswell Park Memorial Institute 1640 media (RPMI) supplemented with 10% (v/v) FBS. Penicillin–streptomycin, L-glutamine and sodium pyruvate were supplemented where necessary. All cells were grown in a humidified incubator at 37 °C with 5% $CO_2$ and sub-cultured when 80 - 90% confluence was reached using 0.25% trypsin in 1 mM EDTA pH 7.4.

**Antibodies**. Primary antibodies used were anti-AXL (CS8661; C44E1), anti-ATM (92356 S), anti-pATM (13050S), anti-ATR (13934 S), anti-pATR (58014S), anti-CHK1 (2360S), anti-pCHK1 (2348S), anti-CHK2 (6334S) and anti-pCHK2 (2197S) from Cell signalling; anti-SAM68 (07-415) and anti-γH2AX (3BW301) from Millipore; antiRPA2 [p Ser33] (NB100-544) from Novus Biologicals; anti-SAM68 (ab76471), anti-SREBP2 (30682), anti-HMGCR (242315) from Abcam; anti-squalene synthase (sc-271602), anti-MVK (sc-390669), anti-OSC (sc-514507), anti-pADPr (sc-56198), anti-GAPDH (sc-47724), anti-ß-actin (sc-47778) from Santa Cruz. Secondary antibodies used were HRP-conjugated goat anti-mouse and goat anti-rabbit from Cell signalling. All primary antibody dilutions were at 1:1,000. Secondary antibody dilutions were at 1:5000.

**Chemicals**. All compounds were reconstituted to 10 mM stock using DMSO (Sigma Aldrich) and stored at −20 °C. AXL inhibitor R428 (BGB324) (#S2841), ATR inhibitor BAY1895344 (#S8666), ATR inhibitor VE-821 (#S8007), PARP1 inhibitor Olaparib (#S1060), PARP1 inhibitor Niraparib (#S2741) were purchased from Selleckchem.

**shRNA knockdown, sgRNA CRISPR–Cas9 mediated KO and lentiviral transduction**. shRNA and sgRNA sequences were obtained from the broad institute genetic perturbation platform (Supplementary Tables 1, 2). shRNA was cloned into pLKO.1 lentiviral plasmid, which was a gift from Bob Weinberg (Addgene plasmid #8453; http://n2t.net/addgene:8453; RRID: Addgene_8453), while sgRNA were cloned into the lentiCRISPR v2 lentiviral plasmid which was a gift from Feng Zhang (Addgene plasmid #52961; http://n2t.net/addgene:52961; RRID: Addgene_52961). SKOV3 and HeyA8 were infected with pLKO.1 and/or lenti-CRISPR v2 lentiviral plasmid and selected using puromycin (1 μg/ml) or sorted for GFP expression. Single clones were generated for CRISPR KO lines.

**Cell fractionation**. Cell fractionation was carried out using PARIS™ Kit (Thermo Fisher Scientific). Briefly, cells were scraped and washed with PBS. Cell pellets were lysed using ice-cold cell fractionation buffer and incubated for 10 min. Cytoplasmic fractionation was obtained by pelleting nuclear fraction at 500×g. Nuclear fraction pellet was lysed using a cell disruption buffer for 10 min. Both fractions were clarified at 13,500 rpm for 15 min at 4 °C

**Co-Immunoprecipitation (Co-IP)**. Cells were lysed in ice-cold co-immunoprecipitation lysis buffer containing 10 mM Tris-HCL (pH7.5), 150 mM NaCl, 0.5% NP40, 0.25% Sodium deoxycholate, 0.5 mM EDTA (pH8.0) supplemented with 100X protease and phosphatase inhibitor (Thermo Fisher Scientific), 50 units/mL benzonase (Millipore) and 2 mM $MgCl_2$ for 45 min at 4 °C. Supernatants were obtained by pelleting cell debris at 13,500 rpm for 15 min at 4 °C. Totally, 500 μg of protein was incubated with 1 μg of antibody and rotated at 4 °C overnight. Following, samples were incubated with 1% (v/v) bovine serum albumin (BSA) (Sigma Aldrich) blocked Pierce™ Protein A/G Agarose (25 L packed bead volume)

(Thermo Fisher Scientific) for 45 min at 4 °C. Beads were washed with a co-immunoprecipitation buffer and bound proteins were released by boiling in a 2× gel loading buffer.

**Immunoblotting**. Cells were lysed in RIPA lysis and extraction buffer (Thermo Fisher Scientific) supplemented with 100× protease and phosphatase inhibitor (Thermo Fisher Scientific) and sonicated. Samples were resolved using 4–12% SDS-polyacrylamide gel electrophoresis (SDS-PAGE) (Invitrogen) and transferred onto polyvinylidene fluoride (PVDF) membrane (Bio-Rad) using Trans-Blot Turbo Transfer System (Bio-Rad) at a constant voltage. The membrane was blocked using either 5% (v/v) BSA in Tris-buffered saline with 0.1% Tween-20 (TBST) or 5% Blotting Grade Blocker Non-Fat Dry Milk (Bio-Rad) prior to incubation with antibodies of interest. Proteins were visualised using SuperSignal® West Dura Extended Duration Substrate (Thermo Fisher Scientific) and imaged using ChemiDoc MP Imaging System (Bio-Rad).

**IF staining**. Cell lines were grown on coverslips, fixed with 4% paraformaldehyde (PFA) and blocked in 5% BSA with 0.1% Triton-X-100. Samples were incubated with primary antibody overnight and subsequently with Alexa Fluor 594 donkey anti-mouse (A21203) or/and Alexa Fluor 647 goat anti-rabbit (A21245) (Life Technologies). Coverslips were mounted using Glycergel mounting medium, visualised using a Leica light sheet microscope (Leica Microsystems), and analysed using ImageJ. Corrected total cell fluorescence (CTCF) was calculated based on the following formula: CTCF = Cell area × cell mean fluorescence − (area of selected cell × mean fluorescence of background readings). Cholesterol was stained using 50 μg/ml filipin III (Sigma-Aldrich) for 1 h and counterstained with propidium iodide.

**Flow cytometry/Edu-labelling**. Cells were treated accordingly and EdU labelling was carried out using Click-iT™ EdU Alexa Fluor™ 647 flow cytometry assay kit as per written in the manufacturer's protocol (Thermo Fisher Scientific). Cells were then analysed using LSRII Cell Analyser (BD). Results were analysed with FlowJo v10.

**Dose–response curve/drug treatment**. Dose–response curve of the AXL inhibitor (R428) was carried out using varying concentrations for 72 h with DMSO as a negative control. Cell viability was quantified using the CellTiter-Glo Luminescent Cell Viability Assay system (Promega) and plotted using GraphPad Prism and $IC_{50}$ was calculated. For combination treatment with AXL inhibitor, a fixed $IC_{20}$ of AXL inhibitor with varying concentrations of selected inhibitor was used.

**Drug combination and dose reduction index analysis**. CI was calculated using the Chou–Talalay equation[21], (CI) = (D)1/(Dx)1 + (D)2/(Dx)2 where (Dx)1 and (Dx)2 represents each drug alone exerting x% inhibition, while (D)1 and (D)2 were concentrations of drugs in combination to elicit the same effect. CI < 1 indicates synergism; CI = 1 indicates an additive effect; and CI > 1 indicates antagonism. The dose-reduction index (DRI) measures the extent or folds of dose reduction when in a combination, at a given effect of inhibition, compared to each drug alone. (DRI) = (Dx)1/(D)1. CI and DRI plots were plotted against Fa, the fraction of cells affected cells or cells killed.

**Cell proliferation assay**. Cells were seeded onto a 96-well Greiner flat-bottomed white plate. The growth rate was quantified in 48 h intervals by measuring ATP levels using CellTiter-Glo Luminescent Cell Viability Assay (Promega) over a duration of 7 days. The proliferation curve was plotted with GraphPad Prism.

**RNA extraction, reverse transcription and quantitative polymerase chain reaction**. RNA extraction was done using TRIzol® Reagent (#15596026), chloroform and RNeasy Mini Kit (Qiagen). Reverse transcription was carried out using High-Capacity cDNA Reverse Transcription Kit (Thermo Fisher Scientific). Following, qPCR was performed using BlitzAmp qPCR system (MiRXES). All reactions were carried out according to the manufacturer's instructions. The cDNA was diluted to 100 ng before use. mRNA levels were measured with gene-specific primers as listed in Supplementary Table 3 and reactions were carried out using QuantStudio™ 5 Real-Time PCR System (Applied Biosystems). Samples were assayed in triplicates with GAPDH as the internal normalisation controls. Relative mRNA expression was determined using the ΔΔCT method.

**RNA sequencing**. Total RNA was extracted, and RNA integrity and concentration were assessed on 2100 Bioanalyzer Instrument (Agilent). Library preparation was performed using TruSeq RNA Library Prep Kit (Illumina) and sequencing was performed on the Illumina HiSeq 4000 System. Relative gene expression changes were filtered based on log2 fold-change >/< 1.5 with a p-value < 0.05.

**Chick CAM model**. Specific pathogen-free fertilised chicken eggs were purchased from JD-SPF Biotech Co. Ltd., washed, and incubated at 37.5 °C with 70% humidity on embryonic day 0 (ED0). Preparation of the CAM was operated within

a biosafety cabinet. A small hole was pierced through the apex of the shell using an 18-gauge needle, and 3–4 ml albumin was removed on ED3. A 1.5 cm² window of the shell was removed to expose the CAM. The window was then covered and sealed with a 3 M™ Tegaderm™ transparent film. On ED7, $0.75 \times 10^6$ SKOV3 cells suspended in 50 μl Matrigel (#354234, Corning) were engrafted into the blood vessel area of CAM. The targeted area of the vessel was gently bruised using a glass rod before inoculation. IC$_{50}$ concentrations of R428, BAY1895344, or a combination of both were administered using a piece of filter paper on ED12 and ED13. Tumour volume was measured using ultrasound (FUJIFILM Sonosite SII using HSL25x transducer) on ED11 and ED14. All CAM experiments were performed in accordance with relevant guidelines and regulations and approved by National Taiwan University.

**Stable isotope labelling by amino acids in cell culture (SILAC) nuclear co-immunoprecipitation (Co-IP) mass spectrometry analysis.** For SILAC labelling, cells were incubated in DMEM or RPMI (-Arg, -Lys) medium containing 10% dialysed FBS supplemented with 42 mg/l $^{13}C_6 ^{15}N_4$ L-arginine and 73 mg/l $^{13}C_6$ $^{15}N_2$ L-lysine (Cambridge Isotope) for DMEM, 84 mg/l $^{13}C_6 ^{15}N_4$ L-arginine and 50 mg/l $^{13}C_6 ^{15}N_2$ L-lysine (Cambridge Isotope) or the corresponding non-labelled amino acids, respectively. Successful SILAC incorporation was verified by in-gel trypsin digestion and MS analysis of heavy input samples to ensure an incorporation rate of >98%.

**SILAC mass spectrometry analysis.** Labelled cell lines were fractionated to obtain the nuclear fraction and Co-IP was performed prior to mass spectrometry analysis. Samples were boiled at 95 °C prior to separation on a 12% NuPAGE Bis–Tris precast gel (Thermo Fisher Scientific) for 10 min at 170 V in 1× MOPS buffer, followed by gel fixation using the Colloidal Blue Staining Kit (Thermo Fisher Scientific). For in-gel digestion, samples were destained in destaining buffer (25 mM ammonium bicarbonate; 50% ethanol), reduced in 10 mM DTT for 1 h at 56 °C followed by alkylation with 55 mM iodoacetamide (Sigma) for 45 min in the dark. Tryptic digestion was performed in a 50 mM ammonium bicarbonate buffer with 2 μg trypsin (Promega) at 37 °C overnight. Peptides were desalted on Stage-Tips and analysed by nanoflow liquid chromatography on an EASY-nLC 1200 system coupled to a Q Exactive HF mass spectrometer (Thermo Fisher Scientific). Peptides were separated on a C18-reversed phase column (25 cm long, 75 μm inner diameter) packed in-house with ReproSil-Pur C18-AQ 1.9 μm resin (Dr Maisch). The column was mounted on an Easy Flex Nano Source and temperature controlled by a column oven (Sonation) at 40 °C. A 105-min gradient from 2 to 40% acetonitrile in 0.5% formic acid at a flow of 225 nl/min was used. The spray voltage was set to 2.2 kV. The Q Exactive HF was operated with a TOP20 MS/MS spectra acquisition method per MS full scan. MS scans were conducted with 60,000 at a maximum injection time of 20 ms and MS/MS scans with 15,000 resolution at a maximum injection time of 50 ms. The raw files were processed with MaxQuant[63] version 1.5.2.8 and searched against the human Uniprot database (95,934 entries) with preset standard settings for SILAC-labelled samples and the re-quantify option was activated. Carbamidomethylation was set as fixed modification while methionine oxidation and protein N-acetylation were considered variable modifications. Search results were filtered with a false discovery rate of 0.01. Known contaminants, protein groups only identified by site, and reverse hits of the MaxQuant results were removed and only proteins were kept that were quantified by SILAC ratios in both 'forward' and 'reverse' samples.

**Cholesterol-related LC-MS analyses.** The following reagents and materials were purchased from the indicated sources: Methanol, isopropanol and ammonium formate, Fisher Chemical; Ultra-high quality water from an Atrium® Pro lab water system, Sartorius; Tricine, acetonitrile and chloroform, Merck; Formic acid and ammonia solution (25%), VWR Chemical.

**Sample preparation for cholesterol-related analyses.** Cells were quenched using ice-cold 150 mM NaCl, collected using a cell scraper and pelleted down. Double lipid extraction was performed for the detection of cholesterol and CE. Briefly, methanol, chloroform and 3.8 mM tricine solution (approximately 1:1:0.5 vol/vol) was used to separate polar metabolites (aqueous fraction) from lipid species (organic fraction). Lipid metabolites in the lipid layer were collected and re-extracted by the addition of chloroform. Lipid layers collected were pooled together and purged with nitrogen gas before storing prior to mass spectrometry run. The whole extraction process was done either on ice or under 4 °C conditions.

**LC-MS profiling of cholesterol.** The lipid extracts were analysed in triplicate using LC–MS analysis of cholesterol was performed using an ACQUITY ultra-performance liquid chromatography (UPLC) system (Waters) interfaced with a high-resolution triple-quadrupole time-of-flight mass spectrometer (Triple-TOF®6600, SCIEX) equipped with an electrospray ionisation (ESI) source, in the positive mode. To ensure mass accuracy of the system throughout the batch acquisition, a calibrant delivery system (CDS) was used to introduce a calibration solution for automated mass calibration of the mass spectrometer. The calibration compound was Reserpine (m/z 609.28066) for positive mode. The MS acquisition

parameters were optimised and set accordingly: ion spray voltage 5500 V, nebuliser gases (GS1 and GS2) 50 psi, curtain gas (CUR) 40 psi, source temperature (TEM) 450 °C, declustering potential (DP) 80 V and collision energy spread (CES) 20 V. Liquid chromatography separation was conducted using an Acquity CSH (C18) column (2.1 mm × 50.0 mm, 1.7 μm, Waters) in gradient elution mode at a flow rate of 0.1 mL/min. The composition of the LC mobile phase is as follows – solvent A comprises methanol: water mixture (4:1) with 5 mM ammonium formate, while solvent B comprises methanol with 5 mM ammonium formate. Cholesterol was separated using the following gradient condition: 0–1 min at 1% B, ramping from 1 to 82.5% B for 1–10 mins, holding at 99% B from 10 to 15 min before equilibration at 1% B from 15–17.2 min. The injection volume of each sample was 2 μL.

**LC-MS profiling of CEs.** The lipid extracts were analysed in triplicate using an ACQUITY UPLC system (Waters) in tandem with a SYNAPT G2-Si High Definition Mass Spectrometry (Waters,). A C18 UPLC column (Acquity UPLC CSH column, 1.0 × 50 mm, 1.7 μm, Waters) was used for separation and the mobile phase comprised of two solvents: 'A' comprising of acetonitrile, methanol and water (2:2:1) with 0.1% acetic acid (Merck) and 0.025% ammonia solution (VWR), and 'B' comprising of isopropanol (Thermo Fisher Scientific) with 0.1% acetic acid and 0.025% ammonia solution. The UPLC programme was performed as follows: the gradient was increased from 50% B to 95% B over 10 mins (flow rate of 0.1 mL/min) before B was further increased to 99% for a 5 min wash at a flow rate of 0.15 mL/min. The column was re-equilibrated for 2.2 min at 1% B (flow rate of 0.1 mL/min). The column temperature was maintained at 45 °C and eluent from the LC system was directed into the MS. Next, high-resolution mass spectrometry was performed in positive ESI mode with a mass range of 100–1800 m/z and a resolution of ≥10,000. Cone and desolvation gas flows were set at 40.0 and 600.0 (L/hour) respectively, with a desolvation temperature of 200 °C. The ESI capillary voltage was 2.0 kV for ionisation. Mass calibration was performed using sodium formate prior to the injection of the samples. Quality control (QC) samples consisting of equal aliquots of each sample were run at regular intervals during the batch LC–MS runs. The lipid extracts were dried under nitrogen gas and reconstituted with Solvent 'B' before LC–MS analysis. The injection volume of each sample was 2 μL.

**LC-MS data processing and analysis.** The raw LC–MS data obtained from the lipid extracts were processed using an XCMS-based peak finding algorithm[64]. The QC samples were used to adjust for instrumental drift and total ion count normalisation was performed. Detected mass peaks were assigned putative metabolite identities by matching the respective masses (<10 ppm error) with the KEGG[65] and Human Metabolome Database[66]. Subsequently, metabolite identities were confirmed based on mass spectral comparison with available metabolite standards or with online mass spectral libraries mzCloud.

**Statistical and reproducibility.** All statistical analyses were performed using GraphPad Prism and data were presented as mean with standard error of the mean unless otherwise stated. Two-tailed unpaired student's $t$-test with Welch's correction was performed to compare differences between two individual groups. A $p$-value of less than 0.05 was defined to be statistically significant and is indicated as follows—ns not significant, $*p < 0.05$, $**p < 0.01$, $***p < 0.001$. The $p$-value significance values are indicated in the relevant figures and figure legends.

**Reporting summary.** Further information on research design is available in the Nature Portfolio Reporting Summary linked to this article.

## Data availability

All data needed to evaluate the conclusions in the paper are present in the paper and/or the Supplementary Materials and Supplementary Data file. The data for RNA-seq are deposited in the GEO database (accession No. GSE233776). The SILAC mass spectrometry proteomics data have been deposited to the ProteomeXchange Consortium via the PRIDE[66] partner repository with the dataset identifier PXD042572. The cholesterol metabolomics data have been deposited to the MetaboLights with the study identifier MTBLS7958. Uncropped and unedited blot images can be found in Supplementary Figs. 7–9. Other source data supporting the findings of this study are available from the corresponding author upon reasonable request.

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

## Acknowledgements

We thank Z. Wang and M. Y. Lee for their valuable advice, R. Ettikan, K.T. Kuay and J. Ye for their technical assistance and members of the RH and TWL Laboratory for their support. This work was supported by the Ministry of Science and Technology Taiwan (108-2320-B-002 -013 -MY3) to R.Y.-J.H. R.Y.-J.H is currently supported by the Yushan Scholar Programme by the Ministry of Education, Taiwan (NTU-110V0402). This research is further supported by the National Medical Research Council, Singapore (OFIRG17may061, OFIRG19nov-0106), National Research Foundation, Singapore (NRF-NRFF2015-04, NRF-CRP22-2019-0003, NRF-CRP23-2019-0004), Agency for Science, Technology and Research, Singapore, and the Singapore Ministry of Education under its Research Centres of Excellence initiative. This work was supported by funding from the Singapore Ministry of Health's National Medical Research Council under its Centre Grant scheme to the National University Cancer Institute and Clinician Scientist Award (NMRC/CSA-INV/0016/2017, to D.S.P.T.) and by funding from the Pangestu Family Foundation Gynaecological Cancer Research Fund.

## Author contributions

X.H.Y. performed and generated experimental data and drafted the manuscript. V.S. performed and generated experimental data for the manuscript revision. X.H.Y and Z.W. designed experiments and interpreted the data. Z.J.C.P. and T.T.Z. performed bioinformatic analyses. D.K. performed the SILAC mass spectrometry analysis. Y.Y.H, K.L.E.P., and Y.S.H. performed cholesterol-related mass spectrometry analyses. Y.-C.C. performed xenograft experiments. X.H.Y. and R.Y.-J.H. conceptualised the research, supervised the study and finalised the manuscript W.L.T. and R.Y.-J.H. supervised the study and obtained funding. D.S.P.T. obtained funding.

## Competing interests

The authors declare no competing interests. Ruby Yun-Ju Huang is an Editorial Board Member for Communications Biology but was not involved in the editorial review of, nor the decision to publish this article.
