## [Peer Review File · Communications Biology]

Reviewers' comments:

Reviewer #1 (Remarks to the Author):

1. Brief summary of the manuscript

The authors of this paper have strategically explored the mechanisms underlying the role of AXL in DNA damage and DNA Damage Response (DDR) in Ovarian cancer. Efforts have been made to develop a rationale to use AXL and DDR inhibitors in the form of potential combination therapy. They have performed experiments on multiple ovarian cancer cells and have generated results using various methodologies and assays including sequencing, pathway analysis, cholesterol-based assays, cell proliferation, protein expression etc.

2. Overall impression of the work

The presented work is impressive and meticulously planned. The paper is technically and conceptually sound. The results are presented using nice graphics that are easy to understand and interpret. This work adds important information to the existing literature on AXL. The exploratory role of cholesterol in preventing DNA damage in response to therapeutic agents provides an additional layer of complexity, especially when making the case of potentially using AXL inhibitors as proposed therapeutics in cancer.

3. Specific comments, with recommendations for addressing each comment

In general, the paper is well-written. The acronym usage needs to be corrected throughout the text. I have attached the annotated file with comments on specific sections and my recommendations on including additional literature in the discussion section.

It would be great to look at the level of ABCA1 as cholesterol efflux protein in response to knocking down SAM68 in cancer cells. But this experiment is not required for publishing this paper. The authors need to keep this in mind for their future investigations.

Reviewer #2 (Remarks to the Author):

Yeo and colleagues demonstrate that AXL mediates DNA damage response in ovarian cancer. In particular, AXL inhibition with R428 resulted in an increase in DNA damage as a single agent. The authors then showed that AXL inhibition enhances response to ATRi. The authors then identified that SAM68 is a binding partner with AXL. Furthermore, AXL and SAM68-deficiency or R428 treatment induced elevated levels of cholesterol and upregulated genes in the cholesterol biosynthesis pathway.

Major comments:

1. R428 is an AXL inhibitor. Can the authors include a western that shows specificity of R428 with pAXL/AXL inhibition with these particular cell lines?

2. The authors utilize SKOV3 and HEYA8 cell lines – neither have a TP53 alteration based on the Domcke paper. TP53 alterations account for up to 96% of the most common subtype HGSC. The authors do not state that they are studying HGSOC, but these results are difficult to translate clinically when HGSOC are the most common histology type. Please address and consider repeating the experiments with an ovarian cancer cell line with a TP53 mutation.

3. Line 288-289: The authors stated that the role of AXL in DNA damage and DDR has been relatively unexplored. Suggestion for the authors to add the paper by Mullen et al. "GAS6/AXL Inhibition Enhances Ovarian Cancer Sensitivity to Chemotherapy and PARP Inhibition through Increased DNA Damage and Enhanced Replication Stress" and to also cite Ramakumar et al "AXL inhibition induces DNA damage and replication stress in Non-small cell lung cancer cells and promotes sensitivity to ATR inhibitors" given this work did show the role of AXL in DNA damage and DDR.

4. The authors demonstrate in Fig 1 and 2 that AXL inhibition alone with R428 can decrease tumor cell proliferation through increasing DNA damage. However, the authors then lead to the combination of ATRi with R428 in Figure 3. It is unclear the reason that Fig 3 does not show the cell viability with R428 alone. Can the authors include this with ATRi vs ATRi + R428?

5. For Fig 3A-C, there appears to be a difference in viability between the treatment groups, can

the authors include significance values?

6. For Fig 4A-b, the authors state that there was a response to olaparib + R428, can significance values be added?

7. Fig 6E-F, can significance values be included?

8. Can the authors better clarify the contribution of AXL in Fig 6E and F. What is the cell viability with Control + R428 and SAMKO + R428?

9. The authors state that supplementation of cholesterol reduced DNA damage induced by AXL inhibition and loss of AXL or SAM68. However, Figure 8E shows gamma-H2Ax expression levels of shSREBF2#2 (1.14) and shSREBPF2 + 1uM R428 (1.46) had a similar increase in gamma-H2Ax expression thus the gamma-H2Ax levels of shSREBF2 + 1uM R428 + 10ug/ul chol may be independent of AXL inhibition. Experiments with AXL rescue would better delineate the role of cholesterol, DNA damage, and AXL. Additionally, Supp Fig 6Q shows an increase in cholesterol concentration with R428. Confirming this with genetic inactivation of AXL would further support this finding.

10. The authors state that LC-MS was performed in R428-treated SKOV3 where both free cholesterol and cholesteryl esters (CEs) were identified (Fig 7G and H). However, Fig 7G does not show any a significant increase in free cholesterol after treatment with R428 compared to DMSO. Can this be clarified?

11. For Fig 8, does addition of cholesterol rescue the proliferation of R428 treated cells or also R428 + ATRi?

12. In vivo experiments would strengthen the conclusions.

Minor comments:

1. Line 75: ..rationale use should be rational use

2. Line 296, the author reported that their observations were in line with previous reports on AXL inhibition induced DNA damage possibly through perturbations of cell cycle progression causing replication stress – the authors should cite Mullen et al.

3. Figure S6F, the figure label for the red curve can be better aligned with SAM68 KO S7.

Reviewer #3 (Remarks to the Author):

In this manuscript the authors have demonstrated additive/synergistic inhibition of DNA-repair mechanisms upon treatment with AXL and ATR inhibitors. AXL and its crosstalk with DNA damage mechanisms is known in previous literature (doi.org/10.3389/fcell.2021.652325). Consequently, studies have been conducted to study AXL and ATR inhibition for sensitizing cancer cells (doi.org/10.1158/1541-7786.MCR-20-0414). Similarly, the authors have investigated the mechanisms and effects of AXL and ATR inhibition in ovarian cancer. The article is well written. The motivation of the study is strong, and data collected is meticulous. The results presented in this study is well within the scope of this Journal.

The data pertaining to interaction of AXL with SAM68 is very intriguing and the findings are novel. Additionally, the mechanistic data between the cholesterol regulation and AXL, is particularly useful as an addition to the breadth of knowledge in this area (doi.org/10.1186/s13041-020-00609-1; doi.org/10.1182/blood.V118.21.2211.2211). The authors can improve their manuscript based on comments below:

1. If the authors have patient-datasets correlating expression of AXL or DNA-repair genes with disease survival, such plots can enhance the overall manuscript. Such data can be added to the introductory results.

2. Have the authors conducted any time-based studies to study the G2/M arrest following inhibition?

3. Figure2B, authors can indicate ROI using arrows in the images.

4. In addition of SREBF2 knockdown, have the authors performed any experiments to actively deplete cholesterol using methyl-b-cyclodextrin and then study the inhibitory effects of AXL/ATR

inhibition? Such data can bolster the arguments for the mechanism.

5. Does the combined AXL/ATR inhibition/ or knockouts affect the invasion/migration characteristics in vitro?

Response to Reviewers' comments

Reviewer #1 (Remarks to the Author):

1. Brief summary of the manuscript

The authors of this paper have strategically explored the mechanisms underlying the role of AXL in DNA damage and DNA Damage Response (DDR) in Ovarian cancer. Efforts have been made to develop a rationale to use AXL and DDR inhibitors in the form of potential combination therapy. They have performed experiments on multiple ovarian cancer cells and have generated results using various methodologies and assays including sequencing, pathway analysis, cholesterol-based assays, cell proliferation, protein expression etc.

2. Overall impression of the work

The presented work is impressive and meticulously planned. The paper is technically and conceptually sound. The results are presented using nice graphics that are easy to understand and interpret. This work adds important information to the existing literature on AXL. The exploratory role of cholesterol in preventing DNA damage in response to therapeutic agents provides an additional layer of complexity, especially when making the case of potentially using AXL inhibitors as proposed therapeutics in cancer.

3. Specific comments, with recommendations for addressing each comment

In general, the paper is well-written. The acronym usage needs to be corrected throughout the text. I have attached the annotated file with comments on specific sections and my recommendations on including additional literature in the discussion section.

It would be great to look at the level of ABCA1 as cholesterol efflux protein in response to knocking down SAM68 in cancer cells. But this experiment is not required for publishing this paper. The authors need to keep this in mind for their future investigations.

Reviewer #2 (Remarks to the Author):

Major comments:

1. Include a western that shows specificity of R428 with pAXL/AXL inhibition with these particular cell lines

We thank reviewer 2 for the comment. Specificity of AXL inhibition using R428 was previously shown in our published paper (Antony et al., 2016) (Figure R1). Upon activation by its ligand Gas6, increased expression of pAXL and its downstream signalling molecule pERK were observed in both SKOV3 and HeyA8 cell lines. This increase was inhibited upon treatment of R428, suggesting the selectivity of R428 in AXL inhibition.

Figure R1: Adapted from Antony et al. (2016) Figure 5B: Western blotting for pAXL, pAKT, and pERK in epithelial ovarian cancer cell lines PEO1 and OVCA429 (Epi-A) or SKOV3 and HeyA8 (Mes), preincubated with DMSO [control (Ctrl)], 2.91 mM R428 (GI50 non-Mes) for an hour, then stimulated with GAS6 for 0.5, 3, or 12 hours.

2. The authors utilize SKOV3 and HEYA8 cell lines – neither have a TP53 alteration based on the Domcke paper. TP53 alterations account for up to 96% of the most common subtype HGSC. The authors do not state that they are studying HGSOC, but these results are difficult to translate clinically when HGSOC are the most common histology type. Please address and consider repeating the experiments with an ovarian cancer cell line with a TP53 mutation.

We thank reviewer 2 for the comment. We are aware of the Domcke paper. However, SKOV3 cells used in this manuscript were purchased from ATCC and carry the TP53 mutation. This was confirmed by Sanger sequencing and was previously in Tan et al. (2013). In addition, TP53 mutation in SKOV3 was also reported by multiple sources (Ikediobi et al., 2006; Langland et al., 2010).

3. Line 288-289: The authors stated that the role of AXL in DNA damage and DDR has been relatively unexplored. Suggestion for the authors to add the paper by Mullen et al. "GAS6/AXL Inhibition Enhances Ovarian Cancer Sensitivity to Chemotherapy and PARP Inhibition through Increased DNA Damage and Enhanced Replication Stress" and to also cite Ramakumar et al "AXL inhibition induces DNA damage and replication stress in Non-small cell lung cancer cells and promotes sensitivity to ATR inhibitors" given this work did show the role of AXL in DNA damage and DDR.

We thank reviewer 2 for the comment. We have now included both the citations in the text. In the revised version of the manuscript, we have also rephrased the sentence as described below:
“The role of AXL in DNA damage and DNA damage response (DDR) has been gradually coming to light (Mullen *et al.*, 2022; Ramkumar *et al.*, 2021).”

- 4. The authors demonstrate in Fig 1 and 2 that AXL inhibition alone with R428 can decrease tumor cell proliferation through increasing DNA damage. However, the authors then lead to the combination of ATRi with R428 in Figure 3. It is unclear the reason that Fig 3 does not show the cell viability with R428 alone. Can the authors include this with ATRi vs ATRi + R428?**

We thank reviewer 2 for the comment. In Fig 3, a fixed IC₂₀ of R428 was used in combination with a varying concentration of ATRi, hence we are unable to plot R428 alone on the same graph. Individual R428 kill curves of all the cell lines can be found at Supp Fig 3B-H. We seek the reviewer’s understanding.

- 5. For Fig 3A-C, there appears to be a difference in viability between the treatment groups, can the authors include significance values?**

We thank reviewer 2 for the comment. We have indicated the significant difference (denoted by *) in the revised Figure 3A-C.

- 6. For Fig 4A-b, the authors state that there was a response to olaparib + R428, can significance values be added?**

We thank reviewer 2 for the comment. We have indicated the significant difference in the revised manuscript Figure 4A,B,D.

- 7. Fig 6E-F, can significance values be included?**

We thank reviewer 2 for the comment. We have indicated the significant difference in the revised manuscript Figure 6E-F.

- 8. Can the authors better clarify the contribution of AXL in Fig 6E and F. What is the cell viability with Control + R428 and SAMKO + R428?**

We thank review 2 for the comment.

Since AXL and SAM68 are interacting, we wanted to determine if the loss of SAM68 could further escalate the DNA damage caused by the combination treatment of AXL and ATR inhibition. To clarify, the control was the combination treatment of R428 + ATRi performed in a control cell line (HeyA8 GFP BAY1895344 (Fig. 6E); Control C1 BAY1895344 (Fig 6F)). This was compared with combination treatment of R428 + ATRi performed in SAM KO cell lines (HeyA8 SAM68-1 BAY1895344 (Fig. 6E); SAM68 KO S7 BAY1895344 (Fig. 6F)). Loss of SAM68 was able to decrease the cell viability, further sensitizing cells to the combination treatment with AXL and ATR inhibition. For further visual clarity, the 4th point concentration of the combination treatments were extracted from Figure 6E & F and represented in Figure R2 below.

As Figure R2 was generated for clarification purposes, it will not be included in the revised manuscript.

Figure R2: Bar chart representing 4th concentration point of the combination treatments extracted from Figure 6E (left) and 6F (right). Data represent mean ± s.e.m; *n* = 3 biologically independent experiments.

- The authors state that supplementation of cholesterol reduced DNA damage induced by AXL inhibition and loss of AXL or SAM68. However, Figure 8E shows gamma-H2Ax expression levels of shSREBF2#2 (1.14) and shSREBPF2 + 1uM R428 (1.46) had a similar increase in gamma-H2Ax expression thus the gamma-H2Ax levels of shSREBF2 +1uM R428 + 10ug/ul chol may be independent of AXL inhibition. Experiments with AXL rescue would better delineate the role of cholesterol, DNA damage, and AXL. Additionally, Supp Fig 6Q shows an increase in cholesterol concentration with R428. Confirming this with genetic inactivation of AXL would further support this finding

We thank reviewer 2 for the comments.

Since previous data suggest that cholesterol upregulation upon AXL inhibition might have a protective role against DNA damage, we wanted to determine whether endogenous cholesterol has any effect on DNA damage. Hence experiments were carried out in shSREBP2/shHMGCR cell lines.

We agree that the loss of SREBP2 also increased γH2AX expression. This could be explained by the fact that cholesterol is an essential component of the cell membrane, and the loss of cholesterol synthesis in shSREBF2 cell lines might lead to increased cell death. However, inhibition of AXL further increased the γH2AX expression by 32%, which was consistent with our hypothesis that AXL inhibition increased DNA damage. Since AXL inhibition was performed in shSREBF2 cell lines, AXL inhibitor-induced upregulation of cholesterol biosynthesis is unlikely due to downregulation of SREBP2, hence leading to further DNA damage. As cholesterol supplementation in R428 treated cells decrease γH2AX levels, these data support the possibility of cholesterol in preventing or reducing DNA damage.

Therefore, to prevent misunderstanding, we have rephrased the sentences as described below:

“This decrease in cholesterol levels correlated with the elevated levels of γH2AX expression in shSREBF2 cell lines (Fig. 8E), which was further enhanced with AXL inhibition (~30% increase) (Fig. 8E). Importantly, supplementation of cholesterol delayed the increase in γH2AX expression induced by AXL inhibition, albeit not to the levels of DMSO.”

The authors state that LC-MS was performed in R428-treated SKOV3 where both free cholesterol and cholesteryl esters (CEs) were identified (Fig 7G and H). However, Fig 7G does not show any a significant increase in free cholesterol after treatment with R428 compared to DMSO. Can this be clarified?

We thank reviewer 2 for the comment. To clarify, cholesterol exists in cells as free cholesterol. When in excess, they are esterified to form cholesteryl ester by the enzyme ACAT (SOAT). When cholesterol levels are low, cholesteryl ester can be converted back to cholesterol. They are interconvertible. Hence no significant increase in free cholesterol after treatment with R428 compared to DMSO was not surprising.

10. For Fig 8, does addition of cholesterol rescue the proliferation of R428 treated cells or also R428 + ATRi?

We thank reviewer 2 for his/her comment. To determine if the addition of cholesterol could reduce the proliferation of cells treated in combination with R428 and ATRi, proliferation assay was carried out using SKOV3 cell line treated with IC₂₀ ATR and combination treatment of IC₂₀ R428 and ATRi for 6 days, with/without supplementation of 10ug/ml cholesterol (Figure R3). A decrease in cell proliferation was observed when SKOV3 cells were treated with ATRi or with the combination treatment. Unlike R428 treated cells, this decrease in cell proliferation could not be rescued by the addition of exogenous cholesterol. This might be due to increased DNA damage to catastrophic levels upon combination treatment (Supp Fig 3N & M).

Figure R3: Proliferation curve of SKOV3 treated with ATRi or combination treatment of R428 and ATRi, supplemented with or without cholesterol. *n* = 3 biologically independent experiments

11. In vivo experiments would strengthen the conclusions

We thank reviewer 2 for the comment. We did perform *in ovo* experiments and the results are shown in Figure 3I. As *in vivo* experiments were pardoned by the editors, we did not further pursue as it was out of scope for this study and we seek your understanding.

Minor comments:

1. Line 75: ..rationale use should be rational use

We thank reviewer 2 for the comment and apologize for our oversight. We have amended the mistake.

2. Line 296, the author reported that their observations were in line with previous reports on AXL inhibition induced DNA damage possibly through perturbations of cell cycle progression causing replication stress – the authors should cite Mullen et al.

We thank reviewer 2 for the comment. We have included the citation in the revised manuscript.

3. Figure S6F, the figure label for the red curve can be better aligned with SAM68 KO S7.

We thank reviewer 2 for the comment. We have rectified the alignment in the revised manuscript figures.

Reviewer #3 (Remarks to the Author):

- If the authors have patient-datasets correlating expression of AXL or DNA-repair genes with disease survival, such plots can enhance the overall manuscript. Such data can be added to the introductory results.**

We thank reviewer 3 for the comment. Based on TCGA dataset, we observed that in ovarian cancer patients that are homologous recombination deficient (HRD), there were no significant difference in the median overall survival of patients with high AXL expression vs patients with low AXL expression (52.63 months VS 54.87 months, HR=1.124 (0.685-1.845), $p=0.643$) (Figure R4A). Similar observations were also observed in patients that are HR proficient (34.87 months VS 38.77 months, HR=1.284 (0.862 - 1.914), $p = 0.218$) (Figure R4A).

The same observation was also showed in the median progression free survival (PFS) where differential AXL expression level no significance difference in either HRD patients (AXL-high 20.63 months VS AXL-low 20.2 months, HR=0.716 (0.463 - 1.110), $p = 0.575$ or HR proficient patients (AXL-high 16.33 months Vs AXL-low 16.66 months, HR=1.044 (0.7201 - 1.513), $p = 0.857$) (Figure R4B). This suggested that the level of AXL expression does not affect the median OS and PFS in either HRD or HR-proficient patients.

Since there was no correlation observed between AXL expression and HR status, we did not include these findings in our results. We seek the understanding of reviewer 3.

Figure R4: (A) Overall survival curve of AXL expression in relation to homologous recombination status (B) Progression free survival curve of AXL expression in relation to homologous recombination status

- Have the authors conducted any time-based studies to study the G2/M arrest following inhibition?**

We thank review 3 for the comment. To evaluate the effects of AXL inhibition on G2/M phase cell cycle arrest in a time-dependent manner, we performed the cell cycle analysis on SKOV3 and HeyA8 cell lines treated with R428 for 16hr, 24hr and 48hr. Cell cycle analysis exhibited an increase in cell population in the S phase at an early time point (16h) upon AXL inhibition, with a subsequent decrease in S phase and increase in G2/M phase at a later time point (24h & 48h) (Figure R5). As replication stress occurs during S phase, the data suggest that AXL

inhibition might increase replication stress which eventually led to a G2/M phase cell cycle arrest.

In the revised version of the manuscript figures, we have amended Fig 1D and Supp Fig 1D with the respective improved results and amended the figure legends accordingly. In addition, we have also amended the manuscript to include this findings as described below:

“Strikingly, cell cycle analysis exhibited an initial increase in S phase followed by G2/M phase upon AXL inhibition, signifying a G2/M phase arrest.”

Figure R5: Quantification of cell cycle analysis of (A) SKOV3 treated with R428 for 16hrs and 24hrs ; and (B) HeyA8 treated with R428 for 16hrs, 24hrs and 48hrs, showing the percentage of cells in different phases of cell cycle (G1, S, G2/M [G2 and mitosis])

3. Figure2B, authors can indicate ROI using arrows in the images.

We thank reviewer 3 for the comment. We have indicated the ROI using arrows in the images. In the revised version of the figure legend, we have also indicated the arrows as described below:

“(B) Immunofluorescence staining of γ H2AX staining (arrows) in DMSO and R428 treated SKOV3. Blue, DAPI; Green, γ H2AX”

4. In addition of SREBF2 knockdown, have the authors performed any experiments to actively deplete cholesterol using methyl-b-cyclodextrin and then study the inhibitory effects of AXL/ATR inhibition? Such data can bolster the arguments for the mechanism.

We thank reviewer 3 for his/her comment. To investigate the effects of ATR inhibition and combination treatment of AXL and ATR inhibition, we treated SKOV3 cell lines using 1mM methyl- β -cyclodextrin (MBCD) for 4 hours, followed by ATRi or combination treatment of AXLi and ATRi.

Upon MBCD pre-treatment, an increase in IC₅₀ was observed when SKOV3 was treated with an ATR inhibitor, or when in combination with R428 (Figure R6A). This suggested that SKOV3 cell lines pre-treated with MBCD became more resistant to AXL inhibition. This was not surprising as our previous publication showed that AXL signalling was disrupted upon MBCD treatment. Treatment of MBCD disrupted the lipid membrane raft, which is important for AXL-RTK cross-talk, and in turn caused an increase in pAXL levels (Figure R7). This increase in pAXL

resulted in decreased sensitivity of SKOV3 to R428 and the combination treatment of R428 and ATRi. This was also reflected in the IC₅₀ of the combination treatment where a 5-fold increase in IC₅₀ was observed when SKOV3 was pre-treated with MBCD.

Upon supplementation of cholesterol, a slight increase in IC₅₀ was observed in both ATR inhibition and the combination treatment of R428 and ATRi (Figure R6B). This implied that supplementation of cholesterol was able to slightly compensate for the disruption of membrane lipids upon MBCD treatment, albeit not to the original state.

Figure R6: (A) Combination treatment of R428 with BAY1895344 in SKOV3 pre-treated with or without MBCD (B) Combination treatment of R428 with BAY1895344 with or without supplementation of 10mg/ul cholesterol in SKOV3 pre-treated with MBCD. Data represent mean ± s.e.m, *n* = 3 biologically independent experiments. MBCD: methyl-β-cyclodextrin

Figure R7: Adapted from Antony et al. (2016) Figure 4B: Western blotting for AXL-RTK cross-talk and pERK in Epi-A-subtype PEO1 and OVCA429 upon stimulation with GAS6

5. Does the combined AXL/ATR inhibition/ or knockouts affect the invasion/migration characteristics in vitro?

We thank reviewer 3 for his/her comment. The function of AXL in invasion and migration has been well studied and has been shown to mediate migration and invasion (Zhu, Wei, & Wei, 2019). Reduced migration and invasion of cells RNA interference knockdown or AXL or inhibition of AXL signalling were also reported in multiple cancers. Despite it being already well-studied, we performed a transwell migration assay to determine the effects of AXL

inhibition, ATR inhibition and their combination treatment on migration in SKOV3 and HeyA8 cell lines (Figure R8). Upon treatment of IC₂₀ R428 or BAY1895344, a decrease in cell density was observed compared to DMSO. This suggested that AXL inhibition or ATR inhibition affected both SKOV3 and HeyA8's ability to migrate, which was in line with what was reported that AXL inhibition inhibits or decreases migration. Upon combination treatment, cell density drastically decreased compared to DMSO and single treatment, implying that combination treatment drastically decreased migration.

Figure R8: Representative images of transwell migration assay of SKOV3 and HeyA8 cell lines treated with DMSO, IC₂₀ R428, IC₂₀ BAY1895344 and combination treatment of IC₂₀ R428 and IC₂₀ BAY1895344.

References

- Antony, J., et al. (2016). The GAS6-AXL signaling network is a mesenchymal (Mes) molecular subtype–specific therapeutic target for ovarian cancer. *Science Signaling*, 9(448), ra97-ra97. doi:10.1126/scisignal.aaf8175
- Tan, T. Z., et al. (2015). CSIOVDB: a microarray gene expression database of epithelial ovarian cancer subtype. *Oncotarget*, 6(41), 43843-43852. doi:10.18632/oncotarget.5983
- Zhu, C., Wei, Y., & Wei, X. (2019). AXL receptor tyrosine kinase as a promising anti-cancer approach: functions, molecular mechanisms and clinical applications. *Molecular cancer*, 18(1), 153. doi:10.1186/s12943-019-1090-3
- Ikediobi, O. N., et al. (2006). Mutation analysis of 24 known cancer genes in the NCI-60 cell line set. *Mol Cancer Ther*, 5(11), 2606-2612. doi:10.1158/1535-7163.Mct-06-0433
- Langland, G. T., et al. (2010). Radiosensitivity profiles from a panel of ovarian cancer cell lines exhibiting genetic alterations in p53 and disparate DNA-dependent protein kinase activities. *Oncol Rep*, 23(4), 1021-1026. doi:10.3892/or_00000728
- Tan, T. Z., et al. (2013). Functional genomics identifies five distinct molecular subtypes with clinical relevance and pathways for growth control in epithelial ovarian cancer. *EMBO Mol Med*, 5(7), 1051-1066. doi:10.1002/emmm.201201823

REVIEWERS' COMMENTS:

Reviewer #3 (Remarks to the Author):

The Authors have answered the questions satisfactorily. The experiments repeated, and results added to the supplementary section is relevant to the questions and the scope of the manuscript. The authors have presented in detail regarding the fundamental concepts of (1) AXL inhibition and downstream effects, and (2) AXL/ATR inhibition and the proposed effects of SAM68 deficiency. The data for the rebuttal can be considered as technically sound.